# Development of a predictive risk model for school readiness at age 3 years using the UK Millennium Cohort Study

Christine Camacho,[1] Viviane S Straatmann,[1,2] Jennie C Day,[1]
David Taylor-Robinson[1]

## ABSTRACT

**Objectives** The aim of this study is to develop a predictive risk model (PRM) for school readiness measured at age 3 years using perinatal and early infancy data.

**Design and participants** This paper describes the development of a PRM. Predictors were identified from the UK Millennium Cohort Study wave 1 data, collected when participants were 9 months old. The outcome was school readiness at age 3 years, measured by the Bracken School Readiness Assessment. Stepwise selection and dominance analysis were used to specify two models. The models were compared by the area under the receiver operating characteristic curve (AUROC) and integrated discrimination improvement (IDI).

**Results** Data were available for 9487 complete cases. At age 3, 11.7% (95% CI 11.0% to 12.3%) of children were not school ready. The variables identified were: parents' Socio-Economic Classification, child's ethnicity, maternal education, income band, sex, household number of children, mother's age, low birth weight, mother's mental health, infant developmental milestones, breastfeeding, parents' employment, housing type. A parsimonious model included the first six listed variables (model 2). The AUROC for model 1 was 0.80 (95% CI 0.78 to 0.81) and 0.78 (95% CI 0.77 to 0.79) for model 2. Model 1 resulted in a small improvement in discrimination (IDI=1.3%, p<0.001).

**Conclusions** Perinatal and infant risk factors predicted school readiness at age three with good discrimination. Social determinants were strong predictors of school readiness. This study demonstrates that school readiness can be predicted by six attributes collected around the time of birth.

[1]Department of Public Health and Policy, University of Liverpool, Liverpool, UK
[2]Aging Research Centre, Karolinska Institute, Stockholm, Sweden

**Correspondence to**
Christine Camacho;
c.camacho@nhs.net

achievement, predict health and social care needs in adults,[4 5] and are associated with long term health outcomes.[6] There has been a growing policy interest in school readiness as a measure of ECD,[7] and school readiness is a key public health indicator in children in the UK. Good school readiness lays a platform for future learning, employment and health.[8 9]

► Use of a large, representative and contemporary cohort study to demonstrate the feasibility of predicting school readiness from data collected in infancy.
► Multiple imputation and bootstrapping were used to evaluate the impact of missing data and internal validity, respectively.
► The main outcome measure, the Bracken School Readiness Assessment, was developed in the USA and is not routinely used in the UK.
► This model was not externally validated, which would have given an indication of generalisability.

School readiness is currently a major focus in England for policy makers, educators and the public health community[10] and national metrics are collected to capture changes over time. In 2017, 29% of children in England were deemed not school ready at the end of their reception year (aged 4–5 years).[11] The percentage of children school ready was nearly 20% higher in the most affluent decile (80% school ready) compared with the most deprived decile (62% school ready) when areas were classified into deciles according to the Index for Multiple Deprivation.[12] In UK policy there has been a focus on demographic factors e.g. maternal age, in targeting early interventions for children.[13] This study will explore the importance of different variables in predicting school readiness.

Previous research has identified a wide range of variables associated with ECD. Predictive risk models (PRMs) are well-established

## INTRODUCTION

Early childhood is a critical time for lifelong physical, social, emotional and cognitive development. A wide range of factors are associated with early cognitive development (ECD).[1] Interventions in the first 3 years of life can improve the trajectory of ECD[2] and deliver the greatest return on investment,[3] yet it is unclear how best to identify children at most risk of delayed ECD, to enable appropriate targeting of interventions.

Cognitive development measures in children are good indicators of later educational

in many clinical disciplines and have more recently been applied to child development. Using PRMs in this context could facilitate targeted early intervention as part of a proportionate universalism approach, which requires universal action with the scale and intensity of interventions proportionate to the level of need.[6] Most models thus far have shown fair or poor discrimination and there have been very few studies in the UK.[14–18] The aim of this study was to develop, for the first time, a PRM for school readiness measured at age 3 years using perinatal and early infancy data from the UK Millennium Cohort Study (MCS).

## METHODS

### Overview

Data from the MCS were used to explore the relationship between the outcome, school readiness and 29 predictor variables using logistic regression analysis. Following univariable analysis to test for unadjusted associations, automated stepwise regression analyses were used to select variables for inclusion in the PRM. Dominance analysis was used to rank and weight included predictors, and integrated discrimination improvement (IDI) was calculated to assess the difference in performance between models. A receiver operator characteristic (ROC) curve was used to evaluate how well the model discriminated school readiness. The area under an ROC curve (AUROC) gives a measure of how well the regression model predicts school readiness at age 3. Traditionally accepted AUROC cut-off points are: 0.9–1=excellent, 0.8–<0.9=good, 0.7–<0.8=fair, 0.6–<0.7=poor, 0.5–<0.6=fail.[19] Multiple imputation was used to assess the impact of missing data in the sample.

### Data source

The PRM was developed and validated using MCS data. The MCS is a nationally representative birth cohort study which recruited 18 550 children born from September 2000 to January 2002, followed up in ongoing data collection waves. The sampling frame was government child benefit records, which had almost universal coverage at the time of sampling. The sample was clustered at the level of electoral ward and stratified to allow over representation of children living in deprived areas and areas with high concentrations of ethnic minorities.[20] Further information about the MCS sample is available in the cohort profile.[21] Data were collected from the main responder (usually mothers) by trained interviewers in participants' homes using a combination of interviews and self-completed questions. All singleton children in the first (aged 9 months) and second (aged 3 years) waves of the MCS with completed data for the outcome and predictors were eligible for inclusion (n=9487).

### Outcome

School readiness was measured using the Bracken School Readiness Assessment (BSRA) which consists of 6 subtests relating to colours, letters, numbers/counting, sizes, comparisons and shapes.[20] The assessment was carried out by interviewers during the second data collection wave when children were aged approximately 3 years old. The BSRA and its predecessors have demonstrated good reliability[22] and validity against other measures and teacher assessments.[23]

The BSRA raw scores were summed and adjusted for age to provide a standardised composite score.[20] Scores were grouped according to cut-offs recommended by Bracken which reflected a 'normative classification' whereby children were categorised as very delayed, delayed, average, advanced or very advanced.[24] We used the same cut-off score as Bracken (mean standardised composite score <85, 1 SD below mean) but collapsed the categories of delayed or very delayed into a single category equivalent to not being school ready. We have dichotomised the outcome 'school readiness' in line with UK policy, and to allow the testing of a PRM using ROC analysis which requires a binary outcome.[25]

### Predictors

Twenty-nine predictor variables were used, which were collected at age 9 months in the first wave of MCS data collection during which data relevant to pregnancy, birth and the perinatal period was captured retrospectively. These were identified from previous research to predict cognitive development and were included in the MCS.[1 2 4 6 26–33] The selected predictor variables were grouped according to the Dahlgren and Whitehead theoretical model[34] of social determinants of health as depicted in figure 1. This model was chosen to provide a framework for categorising predictors to allow analysis of the determinants of ECD.

#### Group 1: demographic and individual factors

Demographic characteristics included child sex, maternal ethnicity, child weight, pre-term birth, mother's age, home language, maternal mental health and child development categorised as shown in box 1.

#### Group 2: lifestyle factors

Self-reported maternal smoking was coded as 'never smoked', 'smoked before pregnancy' and 'smoked during pregnancy'. Maternal alcohol consumption during pregnancy were categorised as 'never or very infrequent', 'occasional', 'regularly' and 'most or everyday'. Breastfeeding duration was grouped as 'never', '1 week or less', '1–6 weeks', '6 weeks – 6 months' and 'over 6 months'.

#### Group 3: social and community networks

The number of children in household was coded as '1', '2–3' or '4+', and being the eldest or only child was recoded as 'yes' or 'no'. The number of parents or carers was either '1' or '2'. Mothers were asked how much time they had spent time in care before the age of 17, this was recoded as 'yes' or 'no' to indicate if they had ever been in care.

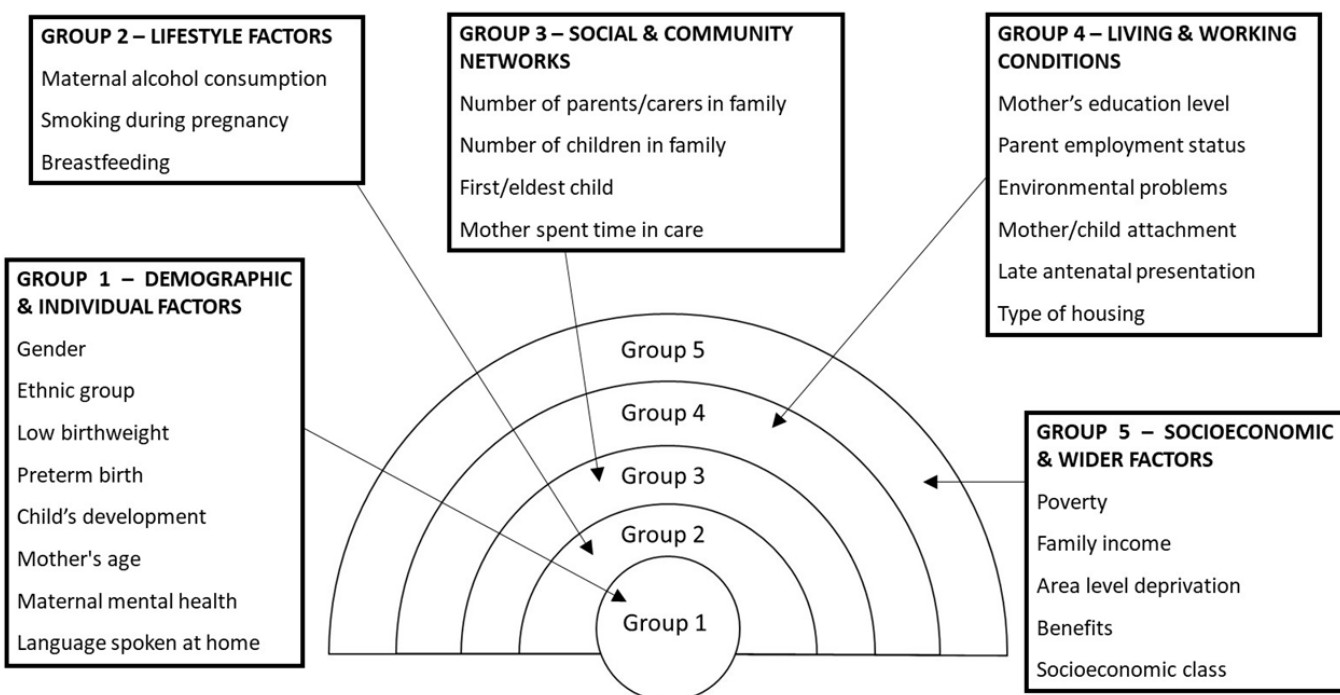

**Figure 1** Rainbow Model showing determinants of school readiness (adapted from Dahlgren and Whitehead[34]).

## Group 4: living and working conditions

Maternal education was categorised into six groups 'degree plus (higher degree and first degree qualifications)', 'diploma (in higher education)', 'A-levels', 'General Certificate of Secondary Education (GCSE) grades A–C', 'GCSE grades D–G' and 'none of these qualifications'. Parent's employment status was classified as either 'both', 'one' or 'neither' parents in work (being on leave from work is classed as being in employment). Housing tenure was coded as 'owner occupied', 'private rented', 'social housing' and 'other'. The response to the question, 'How common is pollution, grime or other environmental problems?' was recoded as 'common', 'not common' and 'not at all'. Presentation for first antenatal visit was recoded as late if after 12 weeks. Maternal attachment was measured using a 6-item Condon Maternal Attachment Questionnaire[35] grouped as 'low (10–21), 'average' (22–23) and 'high (24–27).

### Box 1 Coding of group 1 demographic and individual factors

Categorisation of demographic and individual factors
Child sex: 'female' and 'male'.
Maternal ethnicity: 'white', 'mixed', 'Indian', 'Pakistani and Bangladeshi', 'Black' and 'other'.
Child weight at birth: low (<2.5 kg) or normal/high (≥2.5 kg).
Preterm birth: gestation period less than 37 weeks.
Mother's age in years at birth of first child: grouped into four categories (14–19, 20–29, 30–39, 40+years).
Home language: 'English only', 'English and another language', 'another language only'.
Mental health (1): sad or low for>2 weeks since baby, coded as 'yes' or 'no'.
Mental health (2): diagnosis of depression or serious anxiety, coded as 'yes' or 'no'.
Mental health (3): 9-item modified version of the Rutter Malaise Inventory,[39] coded as 'low' or (0–3) 'high' (4-9) scores.[27]
Child development: – eight items from Denver Developmental Screening Test and five items from MacArthur Communicative Development Inventory, scored on a continuous scale from 13 (above average) to 36 (below average).

## Group 5: socioeconomic and wider factors

The National Statistics Socio-Economic Classification was used to code job details for main respondents (the majority of which were mothers) as: 'managerial & professional', 'intermediate', 'small employers & own account', 'lower supervisory & technical', 'semi-routine & routine', 'never worked & long-term unemployed'. Net household income was reported by identification of the correct band on a show card and grouped into four quartile bands[26] : '£0–£11 000', '£11 000–£22 000', '£22 000–£33 000' and '£33 000+'. Poverty was defined as an equivalised household income 60% below the median before housing costs according to the Organisation for Economic Co-operation and Development Household Equivalence Scale. Families reported receipt of any means-tested benefits, including Jobseekers Allowance, Income Support, Working Families Tax Credit or Disabled Persons Tax Credit. Indices of Multiple Deprivation (IMD) from 2004 which had been retrospectively linked to wave 1 data were used to give small area level deprivation measures.[20] IMD scores were divided into

quintiles, with one the most deprived quintile and five the least deprived.

## Statistical analyses

Analyses were conducted using Stata V.14.2 (StataCorp LP, 2017). Survey weights were applied to take account of clustering, stratification and oversampling in the survey design, and attrition between survey waves, using the svyset command (Pweight=BOVWT2) and svy prefix for regression modelling.[36] The number of events per variable exceeds 35, the predictors were checked for collinearity, a large number of predictors were used and all were significantly associated with the outcome suggesting a robust logistic regression model with sufficient sample size.[37 38]

Descriptive analysis of each predictor and school readiness was carried out to ascertain the prevalence of each predictor in the sample. Univariable logistic regression analyses calculating ORs and 95% CI were carried out to assess the unadjusted association of each variable with the outcome.

A multivariable logistic regression model including all 29 variables was reduced using automated forward and backwards stepwise selection (using a cut-off p value of 0.1). Dominance analysis (repeated regression analyses on subsets of variables) was used to produce a ranking and weighting for each predictor in model 1.[39] These rankings were used to specify a more parsimonious model (model 2) containing the top six predictors, selected to maximise parsimony and performance. The IDI using the complete case sample from model 1 was calculated to assess difference in performance between models as the percentage change in individuals being correctly assigned by the model.[40]

The AUROC and its 95% CI was used to measure discriminatory power of the models. Classification, including sensitivity and specificity, was assessed at the maximised probability cut-off point where the sensitivity and specificity curves intersected. Calibration of the model was assessed using the Pearson $\chi^2$ test.[41] Bootstrapping was used for internal validation of the final model, without repeating selection of predictors in each bootstrap sample. Model performance was assessed using 1000 bootstrap samples, model optimism was averaged across all iterations to obtain an optimism estimate. An optimism-corrected AUROC, which takes account of overfitting, was calculated by subtracting the optimism estimate from the uncorrected AUROC.[42]

A complete case approach was used for the primary analysis. As a sensitivity analysis, multiple imputation by chained equation was performed to impute missing data using the 'mi impute chained' command in Stata. We used predictor variables with relatively little missing data (maternal education, child's sex, mother's age at birth of first child) and the outcome as regular variables in the imputation model. As such individuals with missing data for these four items were not included in the final imputed sample (n=11 897). Twenty imputed datasets were generated, and Rubin's rules were used to calculate results across the imputed datasets.[43]

Robustness tests were carried out in which the final model was tested with an alternative outcome measure for ECD (the British Ability Scales, also tested at age three in the MCS); different coding of outcome and predictor variables (eg, maternal age as a continuous variable); and with the addition of another predictor variable (child care type at age 9 months). See online supplementary file 1 for further details.

## Ethics and patient and public involvement

Ethical approval for each wave of the MCS was granted by NHS Multicentre Research Ethics Committees.[44] No further ethical approval was required for this secondary analysis of MCS data. There was no direct patient or public involvement in this analysis. However, the MCS has an ongoing programme of participant and public engagement.

## RESULTS

There were 15 381 singleton children surveyed in MCS2, of which 13 650 had an outcome recorded for school readiness. Of these children 70% (n=9487) had complete data for the outcomes and all the predictor variables. There were no significant differences in the characteristics of the imputed sample and the complete case sample (p value >0.05 for all $\chi^2$ tests) (table 1); results are reported for complete cases (see online supplementary file 2 for imputed sample results).

About 11.7% (95% CI 11.0% to 12.3%) of children aged 3 years were classified as not being school ready, but this varied significantly by the parents' ethnicity, maternal education and social class (table 1). All 29 predictor variables were significantly associated with school readiness in univariable logistic regression analysis (p<0.1), so none were excluded at this stage.

The stepwise method reduced the final multivariable logistic regression model to 13 predictors: child's sex and ethnicity, mother's age at birth of first child, birth weight, maternal mental health, child development milestones, duration of breastfeeding, number of children in family, maternal education, parents' workforce status, housing tenure, social class and annual family income. In the adjusted analysis, Pakistani and Bangladeshi children were four times more likely to not be school ready than white children (OR 4.19, 95% CI 3.14 to 5.58). The full results are shown in table 2. There was no evidence of collinearity.

Dominance analysis showed that social class was the most important predictor (weighting=17.6), followed by ethnic group (weighting=14.7) and maternal education (weighting=13.8) (table 2). Analysis of the predictor weightings suggests that social factors (average weighting 11.3, SD 4.9) are stronger predictors of school readiness than demographic and lifestyle factors (average weighting 5.5, SD 4.9). IDI was used to test the relative performance of models with all (1-13) variables, with variables added in according to their rank from the dominance analysis (online supplementary file 3). These analyses informed the specification of model 2, which comprised the top

**Table 1** Description of perinatal, sociodemographic and economic characteristics by school ready of sample and imputed sample

| Is child school ready? | Complete cases (n=9487) | | Imputed data (n=11 897) | |
|---|---|---|---|---|
| | Yes (%) | No (%) | Yes (%) | No (%) |
| All | 88.3 | 11.7 | 85.5 | 14.5 |
| **Group 1: demographic and individual factors** | | | | |
| Gender | | | | |
| Female | 91.6 | 8.4 | 89.4 | 10.6 |
| Male | 85.1 | 14.9 | 82.6 | 17.4 |
| Ethnicity | | | | |
| White | 90.4 | 9.6 | 88.6 | 11.4 |
| Mixed | 91.1 | 8.9 | 84.7 | 15.3 |
| Indian | 79.3 | 20.7 | 78.1 | 21.9 |
| Pakistani and Bangladeshi | 55.7 | 44.3 | 56.3 | 43.7 |
| Black or Black British | 79.8 | 20.2 | 68 | 32 |
| Other ethnic group | 73.6 | 26.4 | 74.3 | 25.7 |
| Mother's age at birth of first child | | | | |
| 14–19 | 78 | 22 | 76.4 | 23.6 |
| 20–29 | 87.9 | 12.1 | 86.1 | 13.9 |
| 30–39 | 95 | 5 | 94.4 | 5.6 |
| 40+ | 76.9 | 23.1 | 76 | 24 |
| Birth weight (<2500 g) | | | | |
| Normal/high | 88.8 | 11.2 | 86.1 | 13.9 |
| Low birth weight | 80.2 | 19.8 | 77.7 | 22.3 |
| Maternal mental health (diagnosed depression/anxiety) | | | | |
| No | 89 | 11 | 86 | 14 |
| Yes | 86 | 14 | 84.4 | 15.6 |
| Child developmental milestones | | | | |
| Child development score (mean, 95% CI) | 19.3 (19.2 to 19.3) | 19.9 (19.7 to 20.1) | 19.1 (19.0 to 19.1) | 19.6 (19.4 to 19.7) |
| **Group 2: lifestyle factors** | | | | |
| Duration of breast feeding | | | | |
| 6 months or more | 92.5 | 7.5 | 90.5 | 9.5 |
| 6 weeks–6 months | 89.8 | 10.2 | 87.8 | 12.2 |
| 1–6 weeks | 88.8 | 11.2 | 85.9 | 14.1 |
| 1 week or less | 88.8 | 11.2 | 86.4 | 13.6 |
| Never | 82.6 | 17.4 | 80 | 20 |
| **Group 3: social and community networks** | | | | |
| Number of children in family | | | | |
| One child | 92 | 8 | 89.1 | 10.9 |
| Two or three children | 87.7 | 12.3 | 85 | 15 |
| Four or more children | 71.7 | 28.3 | 70.2 | 29.8 |
| **Group 4: living and working conditions** | | | | |
| Maternal education | | | | |
| Degree plus | 95.6 | 4.4 | 95.1 | 4.9 |
| Diploma | 94.6 | 5.4 | 93.9 | 6.1 |
| A levels | 92.7 | 7.3 | 92 | 8 |
| GCSE A-C | 88.5 | 11.5 | 87.4 | 12.6 |

Continued

**Table 1** Continued

| Is child school ready? | Complete cases (n=9487) | | Imputed data (n=11897) | |
|---|---|---|---|---|
| | Yes (%) | No (%) | Yes (%) | No (%) |
| GCSE D-G | 81 | 19 | 79.1 | 20.9 |
| None | 71.3 | 28.7 | 69.2 | 30.8 |
| Workforce status | | | | |
| Both parents in work | 92.6 | 7.4 | 91.6 | 8.4 |
| One parent in work | 85.8 | 14.2 | 83.4 | 16.6 |
| Neither parent in work | 68.5 | 31.5 | 70.1 | 29.9 |
| Housing tenure | | | | |
| Owner occupied | 91.9 | 8.1 | 90.7 | 9.3 |
| Private rented | 83.8 | 16.2 | 80.5 | 19.5 |
| Social housing | 75.8 | 24.2 | 74.8 | 25.2 |
| Other | 83.4 | 16.6 | 81 | 19 |
| **Group 5: socioeconomic and wider factors** | | | | |
| Social class | | | | |
| Managerial and professional | 95.5 | 4.5 | 94.6 | 5.4 |
| Intermediate | 93.1 | 6.9 | 92.1 | 7.9 |
| Small employers and own account | 91.3 | 8.7 | 89.1 | 10.9 |
| Lower supervisory and technical | 87.2 | 12.8 | 84 | 16 |
| Semiroutine and routine | 81.9 | 18.1 | 80 | 20 |
| Never worked and long-term unemployed | 60.2 | 39.8 | 62.1 | 37.9 |
| Annual income | | | | |
| £33 000+ | 95.7 | 4.3 | 94.9 | 5.1 |
| £22 000–£33 000 | 92.5 | 7.5 | 91.7 | 8.3 |
| £11 000–£22 000 | 85 | 15 | 83.9 | 16.1 |
| £0–£11 000 | 73.8 | 26.2 | 74.1 | 25.9 |

six predictors: social class, child's ethnic group, maternal education, income band, sex and number of children (see online supplementary file 4 for model 2 results).

The AUROC was 0.80 (95% CI 0.78 to 0.81) for model 1 (n=9487), which indicates a 'good' level of discrimination.[19] The AUROC for model 2 (n=11146) was 0.78 (95% CI 0.77 to 0.79). Internal validation using bootstrap optimism correction suggests that the model would have good discriminatory power in an independent sample (adjusted AUROC model 1=0.79, model 2=0.76). The Pearson $\chi^2$ tests were both non-significant indicating adequate calibration (model 1, p=0.07, model 2, p=0.13).[45] IDI showed there was a small but significant difference in performance, with model 1 resulting in a 1.3% (p≤0.001) improvement in discrimination (figure 2).

Sensitivity and specificity were plotted against probability cut-offs to select the optimal cut-off point to assess the PRM's classification (model 1, cut-off=0.12; model 2, cut-off=0.14) (figure 3). For model 1, at this cut-off point sensitivity was 72% (95% CI 69.0% to 74.3%) and specificity was 74% (95% CI 73.5% to 75.3%). Sensitivity of model 2 was similar—72% (95% CI 69.9% to 74.5%). Specificity was lower—71% (95% CI 69.6% to 71.4%),

so this model would generate more false positive results than the model 1, but performance was still in the acceptable range. At a probability cut-off of 12%, 31% of the screened population tested would be identified as being 'at risk' of poor school readiness using model 1.

A sensitivity analysis using an alternative outcome measure (British Ability Scales, BAS), showed that the BSRA measure led to improved discrimination (AUROC=0.79 (95% CI 0.78 to 0.81) for BAS; AUROC=0.80 (95% CI 0.78 to 0.81) for BSRA, p=0.002). See online supplementary file 1 for further details.

## DISCUSSION
### Findings
This study developed a PRM for school readiness at age 3 years using perinatal and early childhood data from the MCS. Model 1 with 13 variables had good discrimination (AUROC=0.80) and classification (sensitivity=72%, specificity=74% at a maximised cut-off). Dominance analysis found the most important variables in predicting school readiness related to socioeconomic conditions (social class, maternal education, family income) and

**Table 2** Unadjusted and adjusted associations and dominance analysis for the predictor variables in model 1 (13 predictors)

| Predictors | Unadjusted OR (95% CI) | Adjusted OR (95% CI) | Weighting (rank) |
|---|---|---|---|
| **Group 1: demographic and individual factors** | | | |
| Gender | | | |
| Female | 1 | 1 | 9.5 (5) |
| Male | 1.76 (1.54 to 2.01) | 2.03 (1.72 to 2.39) | |
| Ethnicity | | | |
| White | 1 | 1 | 14.7 (2) |
| Mixed | 1.4 (0.96 to 2.04) | 1.42 (0.78 to 2.58) | |
| Indian | 1.85 (1.23 to 2.77) | 2.58 (1.65 to 4.03) | |
| Pakistani and Bangladeshi | 5.94 (4.82 to 7.32) | 4.27 (3.20 to 5.69) | |
| Black or Black British | 4.06 (2.90 to 5.69) | 2.1 (1.13 to 3.88) | |
| Other ethnic group | 2.33 (1.38 to 3.93) | 2.92 (1.55 to 5.48) | |
| Mother's age at birth of first child | | | |
| 30–39 | 1 | 1 | 2.9 (11) |
| 40+ | 2.83 (2.29 to 3.49) | 1.05 (0.68 to 1.63) | |
| 20–29 | 5.57 (4.20 to 7.37) | 1.28 (0.98 to 1.66) | |
| 14–19 | 6.02 (4.84 to 7.48) | 1.32 (0.95 to 1.83) | |
| Birth weight (<2500 g) | | | |
| Normal/high | 1 | 1 | 1.4 (12) |
| Low birth weight | 1.7 (1.34 to 2.16) | 1.26 (0.92 to 1.72) | |
| Maternal mental health (diagnosed depression/anxiety) | | | |
| No | 1 | 1 | 0.4 (13) |
| Yes | 1.33 (1.16 to 1.53) | 1.28 (1.07 to 1.53) | |
| Child developmental milestones | | | |
| Developmental score | 1.07 (1.05 to 1.10) | 1.1 (1.07 to 1.14) | 3.9 (11) |
| **Group 2: lifestyle factors** | | | |
| Duration of breast feeding | | | |
| 6 months or more | 1 | 1 | 3.9 (10) |
| 6 weeks–6 months | 1.25 (1.02 to 1.53) | 1.05 (0.81 to 1.36) | |
| 1 week or less | 1.67 (1.34 to 2.09) | 1.19 (0.89 to 1.59) | |
| 1–6 weeks | 1.68 (1.36 to 2.07) | 1.25 (0.96 to 1.65) | |
| Never | 2.74 (2.29 to 3.27) | 1.49 (1.19 to 1.87) | |
| **Group 3: social and community networks** | | | |
| Number of children in family | | | |
| One child | 1 | 1 | 7.8 (6) |
| Two or three children | 1.44 (1.27 to 1.63) | 1.38 (1.15 to 1.66) | |
| Four or more children | 3.71 (3.04 to 4.54) | 2.67 (1.94 to 3.68) | |
| **Group 4: living and working conditions** | | | |
| Maternal education | | | |
| Degree plus | 1 | 1 | 13.6 (3) |
| Diploma | 1.3 (0.93 to 1.81) | 0.81 (0.53 to 1.24) | |
| A levels | 1.66 (1.22 to 2.25) | 1.02 (0.68 to 1.55) | |
| GCSE A-C | 3.02 (2.34 to 3.90) | 1.3 (0.89 to 1.88) | |
| GCSE D-G | 5.55 (4.21 to 7.30) | 1.54 (1.02 to 2.34) | |
| None | 9.62 (7.61 to 12.16) | 1.68 (1.15 to 2.43) | |
| **Workforce status** | | | |

**Table 2** Continued

| Predictors | Unadjusted OR (95% CI) | Adjusted OR (95% CI) | Weighting (rank) |
|---|---|---|---|
| Both parents in work | 1 | 1 | 6.9 (7) |
| One parent in work | 1.79 (1.49 to 2.14) | 0.82 (0.67 to 1.00) | |
| Neither parent in work | 5.39 (4.36 to 6.67) | 1.21 (0.87 to 1.68) | |
| Housing tenure | | | |
| Owner occupied | 1 | 1 | 5.7 (8) |
| Private rented | 2.68 (2.16 to 3.33) | 1.21 (0.87 to 1.67) | |
| Social housing | 3.89 (3.34 to 4.53) | 1.45 (1.16 to 1.81) | |
| Other | 2.65 (2.10 to 3.35) | 0.9 (0.62 to 1.30) | |
| **Group 5: socioeconomic and wider factors** | | | |
| Social class | | | |
| Managerial and professional | 1 | 1 | 17.4 (1) |
| Intermediate | 1.5 (1.19 to 1.89) | 1.06 (0.77 to 1.45) | |
| Small employers and own account | 2.11 (1.44 to 3.08) | 1.41 (0.87 to 2.28) | |
| Lower supervisory and technical | 3.72 (2.76 to 5.00) | 1.65 (1.09 to 2.50) | |
| Semiroutine and routine | 4.99 (4.13 to 6.01) | 1.97 (1.46 to 2.66) | |
| Never worked and long-term unemployed | 12.07 (9.48 to 15.37) | 2.49 (1.69 to 3.66) | |
| Annual income | | | |
| £33 000+ | 1 | 1 | 12.0 (4) |
| £22 000–£33 000 | 1.71 (1.31 to 2.25) | 1.31 (0.96 to 1.79) | |
| £11 000–£22 000 | 3.97 (3.12 to 5.07) | 1.64 (1.22 to 2.22) | |
| £0–£11 000 | 7.7 (6.10 to 9.72) | 2.26 (1.60 to 3.19) | |

ethnicity. A parsimonious model performed similarly well (AUROC=0.78), suggesting it is possible to predict school readiness at age three fairly well using just six variables from the perinatal period and early infancy.

## Comparison with previous studies

The value added of this study is that it is the first UK study to show that school readiness can be predicted with good discrimination with a small number of variables collected in infancy. The predictors of school readiness identified here corroborate previous findings. Male sex, maternal education, income, family composition, parental employment, housing and breastfeeding

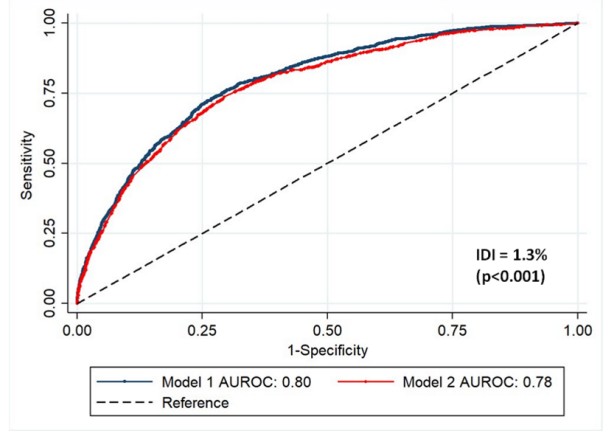

AUROC cut off points are: 0.9–1 = excellent, 0.8–<0.9 = good, 0.7–<0.8 = fair; 0.6–<0.7 = poor, 0.5–<0.6 = fail

**Figure 2** ROC curves for models 1 (13 predictors) and 2 (6 predictors), showing AUROC and IDI. AUROC, area under an ROC curve; IDI, integrated discrimination improvement; ROC, receiver operator characteristic.

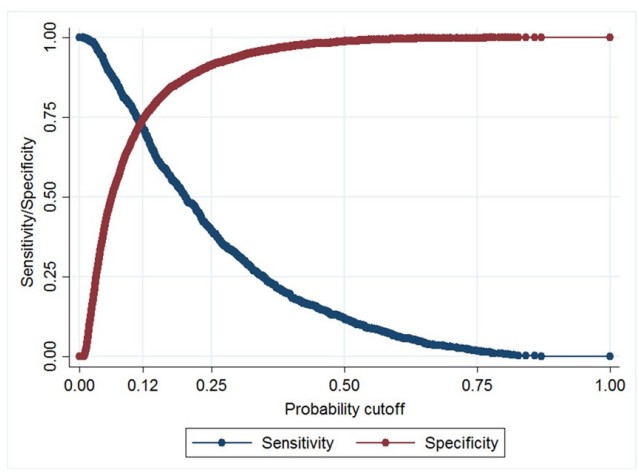

**Figure 3** Maximised probability cut-off of sensitivity and specificity of model 1.

have been identified as significant risk factors of delayed ECD in other studies.[4 14 15 17 18 26] Social factors were the most important predictors, corresponding with current thinking on the social determinants of cognitive development.[6 46]

The model reported here has good predictive strength, and compares favourably to similar PRMs, which with one exception,[17] achieved only fair or poor discrimination.[14 15 18 47] Chittleborough *et al* used the ALSPAC UK birth cohort to test the predictive validity of 2 models for ECD.[14] They used a different outcome measure (School entry assessment aged 4–5) and used six predictors in their model, which appear to be chosen a priori, rather than by a statistical routine. They found that maternal age alone failed to predict ECD (AUROC~0.5), and a model with six predictors achieved only poor discrimination (AUROC=0.67). Camargo-Figuera *et al* used IQ as a measure of ECD and developed a PRM with 12 predictors using the Brazilian Pelotas birth cohort; their model had good discrimination (AUROC=0.8) and calibration, with sensitivity and specificity of 72% and 74%, respectively.[17] We believe the use of a representative cohort for model development, stepwise regression to select predictor variables and dominance analysis to specify a simplified model contributed to the good performance of this PRM.

## Strengths and limitations

A strength of this study was the use of a representative and contemporary UK cohort study as the data source. This offered a wide range of predictor variables and a large sample size which minimised the likelihood of overfitting. The cohort design also ensured correct temporal ordering and blinding with respect to the predictors. A theoretical model informed the PRM and statistical selection was used to specify variables. Multiple imputation was used to assess the impact of missing data. Bootstrapping showed good internal validity.[48]

There are some limitations of this study to be considered. The main outcome, the BSRA, while validated as a measure of school readiness, was developed in the USA and is not routinely used in the UK.[23] The BSRA measures a small set of pre-academic skills and as such is a limited measure of child development, which can be defined as including broader behavioural and social skills. However, an analysis of MCS data linked to teacher reports showed that Bracken scores are strongly associated with the broader Early Years Foundation Stage (EYFS) measure of school readiness used in English schools.[4] The outcome variable was dichotomised to allow ROC curve analysis. We acknowledge the limitations of dichotomising school readiness ethically, conceptually (eg, children develop at different rates) and statistically (ie, loss of information).[49 50] Longitudinal studies are subject to attrition and non-response which can introduce attrition bias, the use of survey weights partially adjust for this, but it was not possible to use these when calculating the AUROC. Sensitivity analysis using multiple imputation showed the effect of missing data was negligible, similar to other

PRMs.[14 15] Most of the predictor variables were based on maternal self-report which may be subject to recall bias, and external validation was not conducted. The predictor variables identified may not be causally associated with school readiness and there are other predictors which may be associated with the outcome which were not included in this model, for example, the home learning environment (which was not assessed at 9 months in the MCS) and childcare in infancy.[51]

## Policy implications

The existing literature, and these findings, indicate that a PRM could plausibly be used to identify a group of children at high risk of poor ECD who may benefit from early intervention. If implemented as part of a 'proportionate universalism' approach,[6] PRMs could mitigate socioeconomic inequalities by providing early years settings with a mechanism for directing their resources to those children at highest risk of poor cognitive development. With new child and maternity datasets now being collected electronically in England, it may be possible to apply a PRM at population level through the use of linked administrative datasets as has been done in Australia.[15]

Poor cognitive development is associated with a range of negative health and social outcomes and contributes to inequalities in society,[3 5 6] so this is of public health importance. Chittleborough *et al* showed that even a model with poor discrimination has benefits over just using young maternal age to direct resources.[14] Similarly, McKean *et al* established that their PRM was better than existing clinical tools used to identify higher-risk children for early intervention.[47]

The practical implications of using such a PRM as a screening tool should be considered. The model reported here would identify 31% of children screened as being 'at risk' of delayed school readiness. An exemplar English Local Authority with a total population of 230 000, and 3000 children aged under 1 year would identify 900 'at risk' children per year if the PRM was applied to this cohort. This percentage equates with national data; in 2015/2016, 31% of children in England were not school ready when tested at age 4–5.[11] However, the overall accuracy of the model is 74%, so over 200 children would be incorrectly classified. PRMs raise ethical issues; labelling very young children as being at risk of poor development could be stigmatising for families, particularly when social factors are the strongest predictors as in this analysis. PRMs would generate false positives (and false negatives), which could cause unnecessary distress and use of resources.

Use of PRMs to identify children at risk of developmental delay should include support and counselling for families, as well as timely access to appropriate interventions. Nelson *et al*[18] comment that Early Intervention services would be overwhelmed by the level of demand generated by such PRMs.[18] A criterion for screening programmes is that interventions should be available, it is thus important to further consider the implications

of using a PRM to assess ECD in the context of available resources. Investment in early intervention would be required, which would have opportunity costs for services locally. Further research is needed to test the external validity of this PRM for example in another cohort or with linked administrative datasets such as the EYFS data from English schools. Alternative modelling approaches which do not require a dichotomous outcome could also be tested. Findings from such models could offer more nuanced predictions on school readiness.

## CONCLUSION

This study has identified a set of predictive risk factors from the perinatal period and early infancy that can predict school readiness at age 3 with a good level of accuracy. Poor cognitive development is socially patterned, evident from a very young age and leads to persistent disadvantage throughout life. It is possible that PRMs could be used to identify high risk children and target appropriate interventions and resources to improve their developmental trajectories, and to reduce social inequalities early in the life course.

**Acknowledgements**  We would like to acknowledge all the families and researchers who are part of the UK Millennium Cohort Study, without whom this research would not have been possible.

**Contributors**  CLC, JCD and DTR planned the study. CLC and VSS conducted the analysis under the supervision of DTR. CLC led the drafting of the manuscript; the research was initially done as part of an MPH. All authors contributed to data interpretation, manuscript drafting and revisions and agreed the submitted version of the manuscript.

**Funding**  This work was supported by the UK Public Health Research Consortium (PHRC). The PHRC is funded by the Department of Health and Social Care Policy Research Programme. Information about the wider programme of the PHRC is available from http://phrc.lshtm.ac.uk/. DTR is funded by the MRC on a Clinician Scientist Fellowship (MR/P008577/1).

**Disclaimer**  The views expressed in this paper are those of the authors and do not necessarily reflect those of the Department of Health and Social Care.

**Competing interests**  None declared.

**Patient consent for publication**  Not required.

**Ethics approval**  Ethical approval for each wave of the MCS was granted by NHS Multicentre Research Ethics Committees.

**Provenance and peer review**  Not commissioned; externally peer reviewed.

**Data sharing statement**  The Millennium Cohort Study dataset is available from the UK Data Service. Millennium Cohort Study: First Survey, 2001-2003: http://doi.org/10.5255/UKDA-SN-4683-4. Millennium Cohort Study: Second Survey, 2003-2005: http://doi.org/10.5255/UKDA-SN-5350-4. The data collection is available to registered or authorised users.

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
