## [Reviewer comments · BMJ Open]

ARTICLE DETAILS

TITLE (PROVISIONAL)	Development of a Predictive Risk Model for School Readiness at age 3 years using the UK Millennium Cohort Study
AUTHORS	Camacho, Christine; Straatmann, Viviane; Day, Jennie; Taylor-Robinson, David

VERSION 1 - REVIEW

REVIEWER	Sharon Wolf University of Pennsylvania
REVIEW RETURNED	21-Jul-2018

GENERAL COMMENTS	Thank you for the opportunity to review the manuscript "Predictive Risk Model for School Readiness at age 3 years using the UK Millennium Cohort Study". This study used data from the UK Millenium Cohort Study to examine how factors early in children's lives predict their school readiness at age 3, and to identify a short list of predictor variables that explain a significant portion of the variance in children being "school ready" or "not school ready". This study has some interesting findings, but there are significant conceptual issues that would need to be addressed for the paper to be suitable for publication. First, the outcome of "school ready" as dichotomized does not seem very useful or accurate. There is no clear definition of school readiness, but what is clear is that it encompasses a set of skills across pre-literacy and pre-numeracy knowledge, behaviors such as self-regulation and attention, executive function, motor skills, and more. As stated in the paper, the Bracken included "colours, letters, numbers, sizes, comparisons and shapes". While these represent pre-literacy and pre-numeracy concepts, I think it is misleading to call this outcome "school ready" vs. "not school ready". A better construct name may be pre-academic skills. Similarly, using this small set of pre-academic indicators to make claims about early cognitive development does not seem reasonable. The specific skills measured in the Bracken have been shown in many studies to be learned in kindergarten (even if children do not attend preschool), and thus seem less important than other potential measures of early cognitive development. Since there are two other measures of child development in the dataset from the Denver Development Screening Test and the MacArthur Communicative Development Inventory, at a minimum the authors should present how much these two measures correlate with the Bracken scores. Why not use of these other two
---

	measures, or a composite of the three, to identify early cognitive development? In terms of the previous literature reviewed, there seem to be other studies that used a similar method to study the same issue. What is the “value added” of this study? The authors can do a better job of explaining what is known, and how their study provides new knowledge. Similarly, in the Discussion section on page 11, the authors note that their findings are somewhat different from previous similar studies. Why might that be, and what do we learn from these new findings? The authors refer to the first round of data used as “at birth”, but it’s actually collected at aged 9-months. While some of the measures ask retrospectively about mothers behaviors while pregnant, this is a bit misleading for some of the outcomes (e.g., when were the other child development measures collected?). Methods Table 1 – can you include statistical tests to show if the differences between the full sample and sample with missing data are statistically different? How were the cut-offs decided for delayed vs. not delayed? It seems the authors used different cut-offs than recommended by Bracken. Because never been validated in UK population, would it be better to use percentile ranks represent the position of a child’s test performance relative to other same age peers who also took the test? Page 5: “Survey weights were applied to take account of clustering, stratification and oversampling in the survey design, and attrition between survey waves”. More details are needed for readers not familiar with the MCS. What was the sampling frame of the MCS and how did this line up with the approach used to account for the nesting / clustering? In general, clearer descriptions of the methods and analytic approach would be helpful for readers not familiar with PRMs.
--	--

REVIEWER	Orla Doyle Associate Professor, School of Economics, University College Dublin, Ireland
REVIEW RETURNED	22-Aug-2018

GENERAL COMMENTS	This paper uses data from wave 1 of the Millennium Cohort Study to predict children’s school readiness at wave 2 and finds that socio-demographic factors are the largest high factors. Overall, this is a good paper, utilising large sample data and applying an interesting method to explore the early predictors of school readiness which may help be useful in identifying children in need to early intervention. I have some questions about the paper which if addressed may improve its clarity. These are listed below. (i) It is not unequivocal that interventions provided in the first 3 years of life always impact children’s developmental trajectories. Therefore I suggest adding the word ‘can’ to the following
--

sentence to reflect this still developing literature: “Interventions in the first three years of life ‘can’ improve the trajectory of ECD”.

(ii) Is the figure stating that 31% of children in the UK are ‘not ready for school’ based on administrative data? How is school readiness measured in this case? Is it similar to the measure used in this paper? It is remarkable that the model here also identifies 31% of children being at risk of poor school readiness.

(iii) It would be useful to explain ‘proportional universalism’ for those not familiar with the term.

(iv) The analysis is based on the Bracken School Readiness Assessment – the paper should state whether this measure is based on parent report or direct assessment by the interviewer. The MSC also includes a measure of the British Ability Scales at age 3, in particular, the Naming Vocabulary scale. The BAS measures seem to be a more frequently used measure than the Bracken scale. How sensitive are the results to using the BAS scale?

(v) The 29 predictor variables were grouped using the Dahlgren and Whitehead model of social determinants of health – as this paper is focused on a cognitive outcome, the authors need to justify why this model is applicable for the present paper.

(vi) Why were the continuous predictor variables turned into binary or categorical variables? Do they need to be in this form for the analysis?

(vii) How were the various cutoffs for the Denver Developmental Screening test, the MacArthur Communicative Development Inventory and the Condon Maternal Attachment scale determined?

(viii) One set of predictors which I would argue is missing from the study are measures of the quality of parenting and the home learning environment. There is only one measure of parenting included – the Condon attachment scale – however there is now a lot of research demonstrating the importance of early parental investment for children’s cognitive and non-cognitive development. I know that there are measures of parental time investment (e.g. how often someone in the home teaches the child to learn the alphabet, count, sing, draw, play sport, etc.) and measures from the Home Observation Measurement of the Environment scale available in the MSC data at age 3, however, there may also be similar measures available at 9 months. In addition, another factor which has been shown to impact children’s cognitive development is childcare. The MSC data includes quite detailed information about childcare at 9 months. I don’t want to publicize my own work but I have a paper with co-authors in which we use the MSC data to examine the causal relationship between the type of childcare at 9 months and school readiness at age 3 and 5 and finds evidence of a relationship e.g.

Cote, S., Doyle, O., Petitclerc, A., Timmins, L. (2013) “Child Care in Infancy and Cognitive Performance Until Middle Childhood in the Millennium Cohort Study”. *Child Development*, 84(4):1191-208.

	(ix) Group 3 Predictors is supposed to capture 'Social and Community Factors', however this set of variables only captures factors within the home such as number of children, number of carers, whether mother spent time in care. Where are the community factors? The MSC data at wave 1 include a number of questions about the respondent's neighbourhood (e.g. satisfaction with neighbourhood, vandalism, rubbish etc.) which could be included. (x) Within the employment categorisation, how is maternity leave coded? (xi) The text states that the Indices of Multiple Deprivation are from 2004, does this mean they were measured at wave 2? It also states that they are used to capture area level deprivation, so are they actually based on the survey data or administrative statistics? (xii) Regarding the main results reported in Table 2, I was somewhat surprised that the wave 1 child development predictors received such a low ranking (2.4) compared to the socio-demographic factors. Usually one of the best predictors of a child's later skills is their early skills, but this doesn't seem to be the case here. What is the correlation between the measures at wave 1 and wave 2? Are the differences due to different aspects of cognitive development being measured? (xiii) I have not used such a PRM and ROC approach before, but is it just a coincidence that the predictors with the highest rankings i.e. ethnicity (6 categories), maternal education (6 categories), social class (6 categories) are all multi-categorical variables, while predictors with binary or less categories receive a lower ranking? (xiv) I really like the robustness test based on multiple imputation reported in the Supplementary File, however it is possible to include the rankings in this table too? (xv) Do you have any explanation for why the results reported here differ from the ones reported in Chittleborough et al. who also use data from the UK? Do they have less/different predictors? (xvi) One of the limitations of this paper, which is not mentioned or addressed, is the issue of unobserved heterogeneity. There may be unobserved child or family factors that influence both the 'predictor' variables and school readiness. Thus the non-causal nature of these associations should be noted.
--	--

REVIEWER	Peter Martin Department of Applied Health Research University College London United Kingdom
REVIEW RETURNED	04-Dec-2018

GENERAL COMMENTS	General comments 1. In many ways this is a carefully designed study that uses sophisticated statistical methods to develop a risk prediction model for children not being school ready on a single large data set. The authors are to be commended for using bootstrap optimism correction, and multiple imputation as a sensitivity analysis.
--

	2. My main concern is a methodological and conceptual one. The authors predict 'school readiness' and discuss its relation to early cognitive development. Conceptually I don't see a reason that either cognitive development or school readiness should be a dichotomous variable. They seem to be continuous variables to me: a child may develop faster or more slowly, they can be more or less well prepared for school (or they could be 'school ready' earlier or later). Methodologically, the outcome, the BSRA, has an interval measurement scale, ranging from 56 to 149 in the Millennium Cohort Study (Connelly 2017: https://cls.ucl.ac.uk/wpcontent/uploads/2017/06/Data-Note-20131_MCS-Test-Scores_Roxanne-Connellyrevised.pdf , p. 15 – also cited by the authors). Categorising this variable to enable logistic regression leads to loss of information (see Connelly 2017, p. 14f). Finally, from the point of view of designing interventions or policies, grouping children into either 'school ready' or 'not school ready' might be less helpful than considering more closely the relative cognitive development of each child. In summary, it seems to me that there are several disadvantages to analysing school readiness as a dichotomous outcome. I would therefore advise the authors to consider whether a statistical model that uses the full information from the interval-level BSRA scale would not serve their purpose better – and if they think it would not, then to justify why not. For methodological reflections on the disadvantages of dichotomising see: Altman, D. (2006). 'The cost of dichotomising continuous variables'. British Medical Journal, 332 (7549), 1080. Senn, S. (2003) Disappointing dichotomies. Pharmaceutical Statistics 2: 239-240. 3. The statistical methods employed are varied and sophisticated. However, there are several errors and omissions in reporting, and in several places it is not possible to ascertain what exactly the authors did. In other places analytical decisions need to be better justified, where currently a justification appears to be lacking or to be incomplete. Please see below for specific comments on these issues. 4. In the interest of Open Science, I strongly encourage the authors to make publicly available the Stata do-files they used to construct their data set and conduct their analysis. This would also help to clarify which specific procedures the authors used (e.g. in bootstrap optimism correction, analysis with weighted data, etc.). Specific comments: Title: The title should make clear that this study reports on the development of a risk model (without external validation), as recommended by #1 of the Tripod checklist. For example: “Development of a predictive risk model for school readiness at age 3 years using the UK Millennium Cohort Study”. Strengths and Limitations:
--	---

Having “a wide range of predictor variables” does not minimise the likelihood of overfitting. I suggest the authors reconsider which true strengths of their study they wish to highlight.

Methods:

p.3 line 54: “Survey weightings were used to correct for attrition and non-response”.

(1) As it stands, this statement appears to be incomplete, as it does not mention weighting for sampling design. I suggest to move this sentence and integrate it with the first paragraph of the Statistical Analysis section, which appears to be more accurate (see also comments on p. 5 line 49).

(2) I advise to refer to “weights”, not weightings.

p. 4 line 3:

Please provide a rationale why only singleton children were included in the study, and consider whether this is likely to affect the generalisability of the findings.

Outcome

It is not clear to me how and why the cut-off point (85) for dichotomising BSRA was chosen. Reference [22] suggests a categorisation, but without justification. Reference [23] does not appear to give any specific justification for dichotomising, or choosing a specific cut-off point. Please clarify on what basis the choice of cut-off point was made.

In fact, the cut-off point seems to lack validity, based on the figures reported by the authors:

according to the authors’ classification, 11.7 % of children were classified as ‘not school ready’ (p. 8), but Public Health England report that 31 % of children were regarded as ‘not school ready’ (p. 3) – almost three times as large a proportion. Given this apparent contradiction, I think the authors need to explain why they believe their cut-off point results in a valid indicator of school readiness.

Predictors

Many continuous predictors were categorised, without justification. It seems to me that this has the potential to weaken the statistical model. Moreover, the cut-off points for categorisation appear arbitrary, since no justification is given for them. Categorisation often leads to loss of information (see the references to Altman and Senn in the general comments above). Why not use the full information from the predictor variables?

p.5, line 33: Why was only mothers’ social class considered as a predictor, and not fathers’?

Statistical analysis

p.5, line 49: Survey weights. Please state the specific Stata command(s) used to weight the data and calculate correct standard errors for statistical analyses on weighted data (e.g. svyset, svy prefix). Please state the name of the weight variable from the MCS data set that was used.

p. 5 line 52ff.

Based on Table 1, the number of children not school ready in the complete cases sample is about $.117 \times 9487 = 1110$. This suggests an EPV of $1110/29 = 38$, not 68 as given in the manuscript. I see no reason to assume that the analysis presented here suffers from sparseness due to too many covariates, or from insufficient EPV. However, the following methodological points are worth considering: In support of their application of the $EPV > 10$ rule of thumb, the authors cite reference Peduzzi et al (1995), which deals with EPV in proportional hazards regression, not logistic regression. I suspect they instead meant to cite: Peduzzi, Peter, et al. "A simulation study of the number of events per variable in logistic regression analysis." *Journal of clinical epidemiology* 49.12 (1996): 1373-1379.

More importantly, the $EPV > 10$ rule has not got a good evidence base, despite Peduzzi et al (1996), which reports on simulations performed on a single data set. I suggest to consider in addition the following, more recent reference, and to revise the manuscript accordingly: Courvoisier, Delphine S., et al. "Performance of logistic regression modeling: beyond the number of events per variable, the role of data structure." *Journal of clinical epidemiology* 64.9 (2011): 993-1000.

p.6, line 17: I suggest the authors consider also reporting the IDI_events and IDI_nonevents (see Pickering JW & Endre ZH (2012) New metrics for assessing diagnostic potential of candidate biomarkers. *Clinical Journal of the American Society of Nephrology* Vol. 7.

p.6, line 28:

Please clarify which method of optimism correction was employed. The cited reference [39] mentions three alternative methods. Also, more detail is required on the bootstrap: what was the number of bootstrap samples drawn? Finally, the "optimised" (optimism-corrected?) AUROC is not 'the difference between the baseline model performance and the performance across the bootstrap samples', as the authors write. A clear short description of bootstrap optimism correction is given in: Austin & Steyerberg (2017) Events per variable and the relative performance of different strategies for estimating the out of sample validity of logistic regression models. *Statistical Methods in Medical Research* 26 (2): 796-808.

p. 6, line 16: While the dominance analysis is sufficiently described, it is not clear to me how the authors decided to select 6 predictors for Model 2 from the 13 contained in Model 1. Why 6 and not 5 or 7, say? What criteria were used precisely? The results section (p. 10) states that IDI was used to assist in selecting the top 6 predictors, but it's not clear from the text or from Supplementary File 2 how the decision was made.

Page 6, line 39ff: The statement on ethical approval, while important, does not belong in the section on Statistical Analysis.

Results

Model 2: The authors should clearly state from the beginning what the top 6 predictors are that are included in Model 2, before statistical comparisons of Model 1 and 2 are described (currently the six predictors are reported in brackets at the bottom of page 10). The authors should add a table displaying the estimated coefficients of Model 2. (Possibly this could be an additional column in Table 2.) An MI sensitivity analysis for Model 2 should be reported in a supplement.

MI data set: The sample size reported for the data set used for multiple imputation varies: it's given as 13,650 in Table 1, but 11,897 in Supplement 1.

p.10, line 41: Why does Model 2 have a different sample size to Model 1? IDI cannot be used to compare models based on different samples (p. 10 line 47).

p.10, line 42: Instead of 'bootstrap optimism', I suggest the term 'bootstrap optimism correction'.

p. 10, line 44: It is not clear whether the adjusted AUROC reported here refers to Model 1 or Model

2. Give the statistic for both models.

Discussion

p. 11 line 27: "...suggesting it is possible to predict school readiness at age 3 using just six variables from the perinatal period and early infancy." I think this conclusion overstates the findings. There are several limitations that I think should receive stronger weight in the conclusions:

- The outcome variable was a measure of school readiness based on an apparently arbitrary cut-off on the BSRA, not an assessment of the child's readiness for school at the point of entering school.
- The outcome variable identified 12 % of children as not school ready, although official data suggest that the actual proportion of 'not-school-ready' children is twice to three times as large. So there are questions about the validity of the outcome used in this study.
- Sensitivity and specificity of the model are rather low, suggesting that practical application of this model would seem to be difficult and subject to many errors of identification. The authors do already allude to some of the negative consequences this could have in practice.

p. 11 line 35: "risk factors of ECD".

Did the authors mean to say "risk factors of delayed cognitive development"?

p. 12 line 9: The authors make too strong a claim regarding what internal validation via bootstrap can demonstrate. Bootstrapping cannot demonstrate generalisability to a different population. We can use the results from bootstrapping to evaluate how likely the results are to be replicated in another sample drawn in the same way from the same population.

	p. 12 line 15: “Many variables were dichotomised or grouped ...”. See my comment under Methods. Since the authors are aware that this is a weakness, why did they choose to go down this route? p. 12, line 48: An average English Local Authority with a population of 230,000 would therefore have 900 ‘at risk’ children per year.” It is not clear how this follows – the authors should describe how they made this calculation. Policy implications The strongest predictors appear to be measures of the parents’ social status: occupational class, ethnicity, income, and education. Were this risk model to be applied in practice, could this mean that children from poorer, less educated, and ethnic minority families would be in danger of being stigmatised as being at risk of ‘not being school ready’? I think the potential social consequences of doing this need to be addressed in the discussion. References Reference 19: The author’s last name is Hansen. Reference 22: The URL does not link to the document. Supplementary File 1: Please use the same reference category for “gender” in the main manuscript and in the MI analysis. Supplementary File 2: I don’t understand what this table is showing me. Is each model compared to the previous one? It would be helpful to have a description of exactly what was done, and which variables are contained in each model. Since there are 13 candidate predictors, wouldn’t the reader expect there to be 13 models to be assessed? Also, I’m not sure how this table helps to select 6 predictors for Model 2 (see also comment on p. 6 line 16).
--	---

VERSION 1 – AUTHOR RESPONSE

Reviewer 1 - Sharon Wolf

1.1. Comment: First, the outcome of “school ready” as dichotomized does not seem very useful or accurate. There is no clear definition of school readiness, but what is clear is that it encompasses a set of skills across pre-literacy and pre-numeracy knowledge, behaviours such as selfregulation and attention, executive function, motor skills, and more. As stated in the paper, the Bracken included “colours, letters, numbers, sizes, comparisons and shapes”. While these represent pre-literacy and pre-numeracy concepts, I think it is mis-leading to call this outcome “school ready” vs. “not school ready”. A better construct name may be pre-academic skills.

Response: School readiness is currently a major focus in the UK public policy context. The aim of this paper is to contribute to this debate by identifying potential areas for targeting of resources to improve school readiness in the children who are likely to need the most support. We have used the term ‘school readiness’ in line with UK policy, in which this outcome is also dichotomised. A dichotomous outcome

is required for ROC analysis, and this modelling approach has been used in many other studies looking at early cognitive development e.g. [1– 4]. We have used a similar methodology to allow comparison with the existing literature and to facilitate pragmatic interpretation for policy makers.

The Bracken School Readiness Assessment (BSRA) has been widely used and is validated for the purpose of assessing whether children are school ready or not [1,2]. We agree that the BSRA encompasses a broad set of skills which may be described or conceptualised by different terminologies in different disciplines and/or national contexts; however as it is a validated measure of school readiness it was felt that to refer to it as such would make the key messages of the paper clear to readers from a variety of backgrounds.

We agree that there are differing definitions of school readiness, however the Bracken School Readiness Assessment (BSRA) has been widely used and is validated for the purpose of assessing whether children are school ready or not [5,6]. The term 'school readiness' is used in a UK policy context and we chose to describe the outcome as such for these reasons. We recognise that this paper is an initial 'high level' analysis in this area, which could lead to targeting of further research including the use of different outcome measures.

We have added some changes to the document to make clearer our rationale for using "school readiness":

"School readiness is currently a major focus in England [10], and national metrics are collected to capture changes over time. In 2017, 29% of children in England were deemed not school ready at the end of their reception year (aged 4-5 years)[11]." p3¹, introduction

"We have used the term 'school readiness' in line with UK policy, in which this outcome is also dichotomised." p4, methods

1.2. Comment: Similarly, using this small set of pre-academic indicators to make claims about early cognitive development does not seem reasonable. The specific skills measured in the Bracken have been shown in many studies to be learned in kindergarten (even if children do not attend preschool), and thus seem less important than other potential measures of early cognitive development.

Since there are two other measures of child development in the dataset from the Denver Development Screening Test (DDST) and the MacArthur Communicative Development Inventory (MCDI), at a minimum the authors should present how much these two measures correlate with the Bracken scores. Why not use of these other two measures, or a composite of the three, to identify early cognitive development?

Response: We recognise the parallels between the construct of school readiness and early cognitive development. Panter & Bracken (2009) and Gredler (1997) distinguish between developmental screening which they categorise as a measure of "potential to acquire new skills" and readiness which assesses skills "related to school learning tasks that are predictive of school success" [7,8]. There is a weak, but statistically significant correlation between the DDST and MCDI and BSRA in this cohort ($r=-0.0662$, $p<0.0001$). The correlation is negative because a higher score on the development test indicated more delayed development. However we were unable to use the child development measures in this analysis because it was necessary that the predictors were collected before the outcome measures and the DDST and MCDI were only collected in wave 1 (i.e. at the same time as the predictor variables) whereas the BSRA was collected later (wave 2).

We acknowledge that there are limitations of the BSRA as a measure of school readiness. In English school, school readiness is based on a teacher-led assessment of achievement against the learning goals of the Early Years Foundation Stage (EYFS) curriculum. The EYFS measure of school readiness is broader than the Bracken School Readiness Assessment (BSRA) however there is evidence that they are strongly associated. An analysis of MCS data linked with the EYFS outcomes showed that

¹ All page numbers refer to the 'Marked Up' version of the main document

Bracken scores aged 3 were the most strongly associated variable with EYFS performance age 5 (OR for being in the bottom decile of EYFS scores: OR=5.8 'very delayed' BSRA score, OR=3.3 'delayed' BSRA, OR=0.4 'above average' BSRA score) [9]. We have added more detail to our discussion on the limitations of the BSRA:

“The main outcome, the BSRA, whilst validated as a measure of school readiness, was developed in the US and is not routinely used in the UK[23]. The BSRA measures a small set of pre-academic skills, but an analysis of MCS data linked to teacher reports showed that Bracken scores are strongly associated with the EYFS measure of school readiness used in English schools [4].” p12-13, discussion

1.3. Comment: In terms of the previous literature reviewed, there seem to be other studies that used a similar method to study the same issue. What is the “value added” of this study? The authors can do a better job of explaining what is known, and how their study provides new knowledge. Similarly, in the Discussion section on page 11, the authors note that their findings are somewhat different from previous similar studies. Why might that be, and what do we learn from these new findings?

Response: The value added of this study is that it is the first UK study to show that school readiness can be predicted with good discrimination with a small number of variables collected in infancy. This could be used by policy makers to assess population need and direct resources.

The introduction has been amended to better reflect what is already known on this subject: “Previous research has identified a wide range of variables associated with early cognitive development. Predictive risk models (PRMs) are well-established in many clinical disciplines and have more recently been applied to child development. Using PRMs in this context could facilitate targeted early intervention as part of a proportionate universalism approach, which requires universal action with the scale and intensity of interventions proportionate to the level of need[6]. Most models thus far have shown fair or poor discrimination and there have been very few studies in the UK [13–17]. The aim of this study was to develop, for the first time, a PRM for school readiness measured at age 3 years using perinatal and early infancy data from the UK Millennium Cohort Study (MCS).” p3, introduction

We have also added some more text to the discussion section to better address what this study adds and why the results are different from other studies:

“Comparison with previous studies

The value added of this study is that it is the first UK study to show that school readiness can be predicted with good discrimination with a small number of variables collected in infancy. The predictors of school readiness identified here corroborate previous findings... The model reported here has good predictive strength, and compares favourably to similar PRMs, ... We believe the use of a representative cohort for model development, stepwise regression to select predictor variables and dominance analysis to specify a simplified model contributed to the good performance of this PRM.” p12-13, discussion

1.4. Comment: The authors refer to the first round of data used as “at birth”, but it’s actually collected at aged 9-months. While some of the measures ask retrospectively about mothers behaviors while pregnant, this is a bit misleading for some of the outcomes (e.g., when were the other child development measures collected?).

Response: Thank you for pointing this out. The 9 month data collection was the first wave of MCS data collection during which data relevant to pregnancy, birth and the perinatal period was collected. Our outcome measure was school readiness at the second wave at child age 3 years. We have clarified this in the text:

“The assessment was carried out by interviewers during the second data collection wave when children were aged approximately 3 years old.” p4, methods

“29 predictor variables were used, which were collected at age 9 months in the first wave of MCS data collection during which data relevant to pregnancy, birth and the perinatal period was captured retrospectively. These were identified from previous research to predict cognitive development and were included in the MCS[1,2,4,6,26–33].” p5, methods

1.5. Comment: Methods. Table 1 – can you include statistical tests to show if the differences between the full sample and sample with missing data are statistically different.

Response: Chi-squared tests have now been added to show the comparison of demographic characteristics of the full sample compared with the sample with missing data. The p value for all the tests was >0.05 indicating no significant differences between the two groups. We have amended the results section to reflect this.

“There were no significant differences in the characteristics of the imputed sample and the complete case sample (p value >0.05 for all chi-squared tests) (Error! Reference source not found.);” p6, results

1.6. Comment: How were the cut-offs decided for delayed vs. not delayed? It seems the authors used different cut-offs than recommended by Bracken. Because never been validated in UK population, would it be better to use percentile ranks represent the position of a child's test performance relative to other same age peers who also took the test?

Response: The cut-offs recommended by Bracken reflected a 'normative classification' whereby children were categorised as very delayed, delayed, average, advanced or very advanced[10]. We used the same cut off score as Bracken (mean standardised composite score [MSCS] <85, which is 1 standard deviation below the mean) but collapsed the categories of delayed or very delayed into a single category equivalent to not being school ready. We have clarified this in the methods section.

“The BSRA raw scores were summed and adjusted for age to provide a standardised composite score[21]. Scores were grouped according to cut-offs recommended by Bracken which reflected a 'normative classification' whereby children were categorised as very delayed, delayed, average, advanced or very advanced [3]. We used the same cut off score as Bracken (mean standardised composite score <85, 1 standard deviation below mean) but collapsed the categories of delayed or very delayed into a single category equivalent to not being school ready.”p4, methods

We carried out a sensitivity analysis using percentile ranks instead of MSCS as the outcome variable which is reported in a supplementary file:

“The BSRA cut off used in the main analysis was a mean standardised composite score <85, which is 1 standard deviation below the mean. The standardisation sample was from a US population. As the BSRA has not been validated in the UK, we tested the model using dichotomised percentile ranks instead of MSCS as the outcome variable (cut off point 1 SD below mean). There was no significant difference in model performance (AUROC=0.80 for both models, p=0.43).” SF1

There is evidence to suggest that within the Millennium Cohort Study percentile scores can be misleading in indicating the difference between the performance of cohort members because they are on an ordinal, rather than interval, scale [3]. We therefore believe that the standardised score is a more reliable outcome measure. We have acknowledged in the discussion that the BSRA has not been validated in the UK.

“The main outcome, the BSRA, whilst validated as a measure of school readiness, was developed in the US and is not routinely used in the UK[24]. The BSRA measures a small set of pre-academic skills, but an analysis of MCS data linked to teacher reports showed that Bracken scores are strongly associated with the EYFS measure of school readiness used in English schools [4].” p13, discussion

1.7. Comment: Page 5: "Survey weights were applied to take account of clustering, stratification and oversampling in the survey design, and attrition between survey waves". More details are needed for readers not familiar with the MCS. What was the sampling frame of the MCS and how did this line up with the approach used to account for the nesting / clustering?

Response: The sampling frame was government child benefit records. Children living in disadvantaged areas and those with high proportions of ethnic minority groups were oversampled, and non-response weights were used to address sample attrition. The MCS sample has been described in detail elsewhere, we have included a reference to the MCS cohort profile [11] and included more detail in the methods section:

"The sampling frame was government child benefit records, which had almost universal coverage at the time of sampling. The sample was clustered at the level of electoral ward and stratified to allow over representation of children living in deprived areas and areas with high concentrations of ethnic minorities[20]. Further information about the MCS sample is available in the cohort profile[21]."p4, methods

In response to this, and another reviewer's comment, we have also moved the information about survey weights to the statistical analysis section, where it is described in more technical detail.

"Survey weights were applied to take account of clustering, stratification and oversampling in the survey design, and attrition between survey waves, using the svyset command (pweight=BOVWT2) and svy prefix for regression modelling[36]"p6, methods

1.8. Comment: In general, clearer descriptions of the methods and analytic approach would be helpful for readers not familiar with PRMs.

Response: Thank you for pointing this out, we would like to paper to be as accessible as possible. We recognise that space is limited, but have suggested the inclusion of an 'overview' paragraph in the methods section to address this comment:

"Data from the MCS were used to explore the relationship between the outcome, school readiness, and 29 predictor variables using logistic regression analysis. Following univariable analysis to test for unadjusted associations, automated stepwise regression analyses were used to select variables for inclusion in the PRM. Dominance analysis was used to rank and weight included predictors, and integrated discrimination improvement (IDI) was calculated to assess the difference in performance between models. A receiver operator characteristic (ROC) curve was used to evaluate how well the model discriminated school readiness. The area under an ROC curve (AUROC) gives a measure of how well the regression model predicts school readiness at age 3. Traditionally accepted AUROC cut off points are: 0.9-1 = excellent, 0.8-<0.9 = good, 0.7-<0.8 = fair, 0.6-<0.7 = poor, 0.5-<0.6 = fail[12]. Multiple imputation was used to assess the impact of missing data in the sample." p3-4, methods

Reviewer 2 - Orla Doyle

2.1. Comment: It is not unequivocal that interventions provided in the first 3 years of life always impact children's developmental trajectories. Therefore I suggest adding the word 'can' to the following sentence to reflect this still developing literature: "Interventions in the first three years of life 'can' improve the trajectory of ECD".

Response: Thank you we agree with this assertion and have changed the wording as suggested. "Interventions in the first three years of life can improve the trajectory of ECD..." p3, introduction

2.2. Comment: Is the figure stating that 31% of children in the UK are 'not ready for school' based on administrative data? How is school readiness measured in this case? Is it similar to the measure used

in this paper? It is remarkable that the model here also identifies 31% of children being at risk of poor school readiness.

Response: Yes, the 31% of children 'not school ready' is based on administrative data from English schools in 2015/16 which was the latest data available at the time of writing. Data for 2016/17 has since been released; the paper has been changed to reflect the most recent data as shown below.

"In 2017, 29% of children in England were deemed not school ready at the end of their reception year." p3, introduction

School readiness in England is measured by a teacher-led assessment of achievement against the learning goals of the Early Years Foundation Stage (EYFS) curriculum. Children are defined as having reached a good level of development if they achieve at least the expected level in the early learning goals in the prime areas of learning (personal, social and emotional development; physical development; and communication and language) and the early learning goals in the specific areas of mathematics and literacy.

The EYFS measure of school readiness is different from the Bracken School Readiness Assessment (BSRA).

"The BSRA measures a small set of pre-academic skills, but an analysis of MCS data linked to teacher reports showed that Bracken scores are strongly associated with the EYFS measure of school readiness used in English schools [4]."p13, discussion

In the MCS sample, 11.7% of children were deemed not school ready at age 3, as measured by the BSRA. Using the probability cut-off which optimises sensitivity and specificity of the model, 31% of the screened population would be identified as being a high risk of poor school readiness. It is coincidental that this percentage was the same as the rate of school readiness in 2016 as measured by administrative data, but this does give an indication that if the model was used at population level it would identify a similar volume of children 'at risk' to those who later identified under the EYFS assessment.

"The model reported here would identify 31% of children screened as being 'at risk' of delayed school readiness. An exemplar English Local Authority with a total population of 230,000, and 3000 children aged under 1 year would identify 900 'at risk' children per year if the PRM was applied to this cohort. This percentage equates with national data; in 2015/16, 31% of children in England were not school ready when tested at age 4-5[11]. However, the overall accuracy of the model is 74%, so over 200 children would be incorrectly classified; this could lead to stigmatisation of families and unnecessary use of resources."p14, discussion

2.3. Comment: It would be useful to explain 'proportional universalism' for those not familiar with the term.

Response: Thank you for suggesting this, we have added an explanation of this term as follows:

"Using PRMs in this context could facilitate targeted early intervention as part of a proportionate universalism approach, which requires universal action with the scale and intensity of interventions proportionate to the level of need [6]." p3, introduction

2.4. Comment: The analysis is based on the Bracken School Readiness Assessment – the paper should state whether this measure is based on parent report or direct assessment by the interviewer. The MSC also includes a measure of the British Ability Scales at age 3, in particular, the Naming Vocabulary scale. The BAS measures seem to be a more frequently used measure than the Bracken scale. How sensitive are the results to using the BAS scale?

Response: The BSRA was directly assessed by the interview. We have now included this sentence:

“The assessment was carried out by interviewers during the second data collection wave when children were aged approximately 3 years old.” p4, methods

Thank you for raising the question of the BAS measure. We have done some sensitivity analyses using this. There is a moderate positive correlation between BAS and BSRA scores ($r=0.5722$, $p<0.0001$). Using BAS as the outcome measure (dichotomised to 1 SD below the mean as cut off for ‘fail’), the result of the predictive risk model is similar; AUROC = 0.79 (95% CI 0.78-0.81) for BAS vs AUROC = 0.80 (95% CI 0.78-0.81) for BSRA. We added the following information:

“Robustness tests were carried out in which the final model was tested with an alternative outcome measure for early cognitive development (the British Ability Scales, also tested at age 3 in the MCS), different coding of outcome and predictor variables (e.g. maternal age as a continuous variable) and the addition of another predictor variable (child care type at age 9 months). See supplementary file 1 for further details.” p7, methods

“An alternative measure of early cognitive development contained in the MSC are the British Ability Scales (BAS), measured at age 3. BAS scores were dichotomised to 1 SD below the mean as cut off for ‘fail’. There is a moderate positive correlation between BAS and BSRA scores ($r=0.5722$, $p<0.0001$). The table below compares performance of the models; there is a small but statistically significant improvement in discrimination using BSRA as an outcome measure compared to BAS.” SF1

Outcome variable	N	AUROC (95% CI)
BSRA	9487	0.80 (0.78,0.81)
BAS	9487	0.79 (0.77,0.80)

Ho: $\text{area}(xb1) = \text{area}(xb6)$; $\chi^2(1) = 9.20$, $\text{Prob}>\chi^2 = 0.002$

“A sensitivity analysis using an alternative outcome measure (British Ability Scales, BAS), showed that the BSRA measure led to improved discrimination (AUROC = 0.79 (95% CI 0.78-0.81) for BAS; AUROC = 0.80 (95% CI 0.78-0.81) for BSRA, $p=0.002$).

See supplementary file 1 for further details.” p12, results

2.5. Comment: The 29 predictor variables were grouped using the Dahlgren and Whitehead model of social determinants of health – as this paper is focused on a cognitive outcome, the authors need to justify why this model is applicable for the present paper.

Response: We agree with the referee that use of the Dahlgren and Whitehead model has not been fully justified. It is evident from school readiness data that there is a social gradient in outcomes [13]. The Dahlgren and Whitehead model has been used in an inequalities context and as a framework for grouping determinants of health [14]. Other PRM studies looking at cognitive development concluded that social rather than biological predictors were more important e.g. [3,4]. We included the model as a framework which allowed predictors to be grouped. There was no a priori assumption that the model would predict the determinants of early cognitive development. We have included the below statements in the introduction and methods section to give this context more clearly.

“There was nearly a 20% point gap in performance between the most (62% school ready) and the least (80%) deprived deciles of Index of Multiple Deprivation [11]. In UK policy there has been a focus on demographic factors e.g. maternal age, in targeting early interventions for children[12]. This study will explore the importance of different variables in predicting school readiness.” p3, introduction

“This model was chosen to provide a framework for categorising predictors to allow analysis of the determinants of early cognitive development.” p5, methods

2.6. Comment: Why were the continuous predictor variables turned into binary or categorical variables? Do they need to be in this form for the analysis?

Response: Many of the predictors were already in a categorical format e.g. social class, housing type, employment status, gender. The continuous variables we specifically recoded were maternal age and the child development scores. The predictors don't need to in this form for the analysis, but we thought it would make interpretation of the results easier. For example, there is a U-shaped relationship between school readiness and maternal age, this information is visible in the odds ratios when age is categorised, but not when it is a continuous variable.

We have carried out sensitivity analyses on the impact of categorising maternal age and developmental test scores, which is summarised in supplementary file 1:

“As a sensitivity analysis, the coding of 4 predictor variables was altered: maternal age (from categorical to continuous), developmental scores (from categorical to continuous) and ethnicity (from categorical to binary). The impact of this on final model performance is shown below:

Description	n	AUROC	Comparative AUROC (n=9310)
Final model	9487	0.80	0.79 (0.77,0.81)
Developmental score (continuous)	9487	0.80	0.80 (0.78,0.81)
Maternal age (continuous)	9310	0.79	0.79 (0.78,0.81)
Ethnicity (binary)	9487	0.79	0.79 (0.78,0.80)

Ho: area(xb1) = area(xb2) = area(xb3) = area(xb4); chi2(3) = 9.98; Prob>chi2 = 0.02

In summary, there were small but statistically significant differences between the models.

The only change which improved model discrimination was using continuous developmental scores, so this was incorporated into the final model. There is a U-shaped relationship between school readiness and maternal age, so there was a clear rationale for including this as a categorical predictor.” SF1

There is a less clear rationale for categorising the developmental test scores, and including this as a continuous outcome led to a slight improvement in model performance, so we have changed this to a continuous variable (see amended text below).

“Child development – 8 items from Denver Developmental Screening Test and 5 items from MacArthur Communicative Development Inventory, scored on a continuous scale from 13 (above average) to 36 (below average)” p5, methods

2.7. Comment: How were the various cutoffs for the Denver Developmental Screening test, the MacArthur Communicative Development Inventory and the Condon Maternal Attachment scale determined?

Response: There was no theoretical basis for these cut offs. The Stata egen cut() function was used to divide the data into 3 roughly equally sized groups for both these predictors. As per the response to comment 2.6 we have used a continuous measure for the developmental test scores in this revised version.

2.8. Comment: One set of predictors which I would argue is missing from the study are measures of the quality of parenting and the home learning environment. There is only one measure of parenting included – the Condon attachment scale – however there is now a lot of research demonstrating the importance of early parental investment for children’s cognitive and noncognitive development. I know that there are measures of parental time investment (e.g. how often someone in the home teaches the child to learn the alphabet, count, sing, draw, play sport, etc.) and measures from the Home Observation Measurement of the Environment scale available in the MSC data at age 3, however, there

may also be similar measures available at 9 months. In addition, another factor which has been shown to impact children's cognitive development is childcare. The MSC data includes quite detailed information about childcare at 9 months. I don't want to publicize my own work but I have a paper with co-authors in which we use the MSC data to examine the causal relationship between the type of childcare at 9

months and school readiness at age 3 and 5 and finds evidence of a relationship e.g. Cote, S., Doyle, O., Petitclerc, A., Timmins, L. (2013) "Child Care in Infancy and Cognitive Performance Until Middle Childhood in the Millennium Cohort Study". *Child Development*, 84(4):1191-208.

Response: Thank you for raising this point, we acknowledge that there are other measures in the MCS which could have been used in this analysis. We have done a sensitivity analysis adding childcare type at 9 months to the final model, which is reported in supplementary file 1:

"Robustness tests were carried out in which the final model was tested with an alternative outcome measure for early cognitive development (the British Ability Scales, also tested at age 3 in the MCS); different coding of outcome and predictor variables (e.g. maternal age as a continuous variable); and with the addition of another predictor variable (child care type at age 9 months). See supplementary file 1 for further details." p7, methods

"There are other measures in the MCS which could have been used as predictors in this analysis.

We have done a sensitivity analysis adding childcare type at 9 months to the final model. This reduces the overall discrimination of the model (AUROC = 0.77 vs 0.80), however this could be due to missing data as the child care variable is less complete. There is a statistically significant association with school readiness and child care type in the multivariable model, with children in formal child care settings more likely to be school ready than those being looked after by parents (OR = 1.76, p=0.02)" SF1

We have included an additional limitation of the analysis that there are other predictors which have not been included and quoted Cote et al (2013) as one example of this.

"...there are other predictors which may be associated with the outcome which were not included in this model e.g. childcare in infancy[15]".p12, discussion

2.9. Comment: Group 3 Predictors is supposed to capture 'Social and Community Factors', however this set of variables only captures factors within the home such as number of children, number of carers, whether mother spent time in care. Where are the community factors? The MSC data at wave 1 include a number of questions about the respondent's neighbourhood (e.g. satisfaction with neighbourhood, vandalism, rubbish etc.) which could be included.

Response: Thanks – to clarify, the response to the question on conditions in the neighbourhood was included in group 4 – Living and working conditions. This asked, "How common is pollution, grime or other environmental problems in your local area?"

"Group 4 – Living and Working Conditions Maternal education was categorised into six groups 'degree plus (higher degree and first degree qualifications)', 'diploma (in higher education)', 'A-levels', 'GCSE grades A–C', 'GCSE grades D–G' and 'none of these qualifications'. Parent's employment status was classified as either 'both', 'one' or 'neither' parents in work². Housing tenure was coded as 'owner occupied', 'private rented', 'social housing' and 'other'. The response to the question, "How common is pollution, grime or other environmental problems?" was recorded as 'common', 'not common' and 'not at all'." p6, methods

There was no significant association with this predictor in the multivariable analysis. However, we acknowledge that there are other variables which could be included relating to community factors. This has been included as a limitation.

² Being on leave from work is classed as being in employment

“...there are other predictors which may be associated with the outcome which were not included in this model...” p12, discussion

2.10. Comment: Within the employment categorisation, how is maternity leave coded?

Response: Being on leave from work is classed as being in employment[16]. A footnote on p6, methods have been added to this effect.

2.11. Comment: The text states that the Indices of Multiple Deprivation (IMD) are from 2004, does this mean they were measured at wave 2? It also states that they are used to capture area level deprivation, so are they actually based on the survey data or administrative statistics?

Response: Thanks. Baseline wave 1 data was linked to small area level measures of deprivation derived from census data. As it happens the 2004 measures of area deprivation were linked retrospectively to baseline data and are provided within the MCS dataset. We have added a note to make this clear:

“Indices of Multiple Deprivation (IMD) from 2004 were linked retrospectively to wave 1 data to give small area level deprivation measure. IMD scores were divided into quintiles, with 1 the most deprived quintile, and 5 the least deprived.” p6, methods

2.12. Comment: Regarding the main results reported in Table 2, I was somewhat surprised that the wave 1 child development predictors received such a low ranking (2.4) compared to the socio-demographic factors. Usually one of the best predictors of a child’s later skills is their early skills, but this doesn’t seem to be the case here. What is the correlation between the measures at wave 1 and wave 2? Are the differences due to different aspects of cognitive development being measured?

Response: There is a weak, but statistically significant correlation between these child development tests used in wave 1 (DDST and MCDI) and BSRA which was tested at wave 2 ($r=-0.0662$, $p<0.0001$). The correlation is negative because a higher score on the development test indicated more delayed development. The content of the tests is quite different. Many of the wave 1 child development measures assess achievement of gross motor milestones such as sitting, walking and picking up an object. The BSRA tests cognitive skills such as shape and colour recognition.

2.13. Comment: I have not used such a PRM and ROC approach before, but is it just a coincidence that the predictors with the highest rankings i.e. ethnicity (6 categories), maternal education (6 categories), social class (6 categories) are all multi-categorical variables, while predictors with binary or less categories receive a lower ranking?

Response: We do not believe there is an association between the number of categories within a predictor variable and the ranking from dominance analysis. We have re-run this analysis using ethnicity as a binary variable and this still ranked highly. Other variables such as breastfeeding duration and maternal age also had multiple categories and were not highly ranked.

2.14. Comment: I really like the robustness test based on multiple imputation reported in the Supplementary File, however it is possible to include the rankings in this table too?

Response: Thank you. Yes, dominance analysis has been done using the multiple imputed data as well now, the results are included in the supplementary file 2 as shown below.

Predictor	Adjusted OR (95% CI)	Weighting (rank)
GROUP 1 - DEMOGRAPHIC & INDIVIDUAL FACTORS		
Gender		
Male	1	8.5 (5)
Female	0.47 (0.41-0.54)	
Ethnicity		
White	1	15.7 (3)
Mixed	1.04 (0.62-1.75)	
Indian	2.68 (1.85-3.89)	
Pakistani and Bangladeshi	3.85 (2.94-5.04)	
Black or Black British	2.31 (1.43-3.72)	
Other ethnic group	3.95 (2.30-6.77)	
Mother's age at birth of first child		
30-39	1	1.5 (12)
40+	1.05 (0.67-1.64)	
20-29	1.22 (0.99-1.51)	
14-19	1.22 (0.93-1.59)	
Birth weight (<2500grams)		
Normal/high	1	
Low birthweight	1.52 (1.18-1.97)	
Maternal Mental Health (Diagnosed depression/anxiety)		1.2 (13)
No	1	1.5 (11)
Yes	1.15 (0.98-1.34)	
Child developmental milestones		
Developmental score	1.10 (1.07,1.13)	2.8 (10)
GROUP 2 - LIFESTYLE FACTORS		
Duration of breastfeeding		
6 months or more	1	3.6 (9)
6 weeks - 6 months	1.17 (0.92-1.48)	
One week or less	1.15 (0.90-1.48)	
1 - 6 weeks	1.22 (0.96-1.57)	
Never	1.58 (1.29-1.95)	
GROUP 3 - SOCIAL & COMMUNITY NETWORKS		
Number of children in family		
One child	1	7.1 (6)
Two or three children	1.40 (1.19-1.63)	
Four or more children	2.48 (1.94-3.16)	
GROUP 4 - LIVING & WORKING CONDITIONS		

Maternal education		
Degree plus	1	
Diploma	0.88 (0.61-1.26)	
A levels	1.13 (0.80-1.59)	
GCSE A-C	1.34 (1.01-1.78)	16.7 (2)
GCSE D-G	1.72 (1.23-2.39)	
None	1.74 (1.28-2.38)	
Workforce status		
Both parents in work	1	
One parent in work	0.94 (0.78-1.12)	6.5 (7)
Neither parent in work	1.21 (0.93-1.57)	
Housing tenure		
Owner occupied	1	
Private rented	1.18 (0.90-1.54)	5.5 (8)
Social housing	1.43 (1.18-1.72)	
Other	0.96 (0.69-1.35)	
GROUP 5 - SOCIOECONOMIC AND WIDER FACTORS		
Social class		
Managerial & professional	1	
Intermediate	0.98 (0.75-1.29)	
Small employers & own account	1.32 (0.87-2.00)	17.6 (1)
Lower supervisory & technical	1.50 (1.06-2.13)	
Semi-routine & routine	1.77 (1.38-2.27)	
Never worked & long-term unemployed	2.19 (1.53-3.15)	
Annual income		
£33,000+	1	
£22,000-£33,000	1.33 (1.02-1.72)	
£11,000-£22,000	1.67 (1.30-2.14)	11.9 (4)
£0-£11,000	2.14 (1.60-2.87)	
ROC Analysis		AUROC = 0.79 (95% CI 0.78 - 0.80)

2.15. Comment: Do you have any explanation for why the results reported here differ from the ones reported in Chittleborough et al. who also use data from the UK? Do they have less/different predictors?

Response: Chittleborough et al (2011) use data from the Avon Longitudinal Study of Parents and Children (ALSPAC) for their analysis [1]. They used a different outcome measure (School entry assessment aged 4-5) and used 6 predictors in their main model. The predictors were chosen a priori, rather than by a statistical routine, and some were not significantly associated with the outcome. We have added some more text to the discussion section to better address what this study adds and why the results are different from other studies:

“Comparison with previous studies

The value added of this study is that it is the first UK study to show that school readiness can be predicted with good discrimination with a small number of variables collected in infancy. The predictors of school readiness identified here corroborate previous findings... The model reported here has good predictive strength, and compares favourably to similar PRMs, ... We believe the use

of a representative cohort for model development, stepwise regression to select predictor variables and dominance analysis to specify a simplified model contributed to the good performance of this PRM.”p12-13, discussion

“Chittleborough et al used the ALSPAC UK birth cohort to test the predictive validity of 2 models for ECD[14]. They used a different outcome measure (School entry assessment aged 4-5) and used 6 predictors in their model, which appear to be chosen a priori, rather than by a statistical routine. They found that maternal age alone failed to predict ECD (AUROC~0.5), and a model with 6 predictors achieved only poor discrimination (AUROC=0.67).” p13, discussion

2.16. Comment: One of the limitations of this paper, which is not mentioned or addressed, is the issue of unobserved heterogeneity. There may be unobserved child or family factors that influence both the ‘predictor’ variables and school readiness. Thus the non-causal nature of these associations should be noted.

Response: Thank you pointing this out. PRM aim to identify predictors of an outcome rather than identify causes, and it is useful to make this explicit. We have added this text to the discussion:

“The predictor variables identified may not be causally associated with school readiness...”p13, discussion

Reviewer 3 - Peter Martin

3.1. Comment: In many ways this is a carefully designed study that uses sophisticated statistical methods to develop a risk prediction model for children not being school ready on a single large data set. The authors are to be commended for using bootstrap optimism correction, and multiple imputation as a sensitivity analysis.

Response: Thank you for this positive feedback.

3.2. Comment: My main concern is a methodological and conceptual one. The authors predict ‘school readiness’ and discuss its relation to early cognitive development. Conceptually I don’t see a reason that either cognitive development or school readiness should be a dichotomous variable. They seem to be continuous variables to me: a child may develop faster or more slowly, they can be more or less well prepared for school (or they could be ‘school ready’ earlier or later).

Methodologically, the outcome, the BSRA, has an interval measurement scale, ranging from 56 to 149 in the Millennium Cohort Study (Connelly 2017: https://cls.ucl.ac.uk/wpcontent/uploads/2017/06/Data-Note-20131_MCS-Test-Scores_Roxanne-Connelly-revised.pdf , p. 15 – also cited by the authors). Categorising this variable to enable logistic regression leads to loss of information (see Connelly 2017, p. 14f).

Finally, from the point of view of designing interventions or policies, grouping children into either ‘school ready’ or ‘not school ready’ might be less helpful than considering more closely the relative cognitive development of each child.

In summary, it seems to me that there are several disadvantages to analysing school readiness as a dichotomous outcome. I would therefore advise the authors to consider whether a statistical model that uses the full information from the interval-level BSRA scale would not serve their purpose better – and if they think it would not, then to justify why not.

For methodological reflections on the disadvantages of dichotomising see:

Altman, D. (2006). 'The cost of dichotomising continuous variables'. *British Medical Journal*, 332 (7549), 1080.

Senn, S. (2003) Disappointing dichotomies. *Pharmaceutical Statistics* 2: 239-240.

Response: Thanks – as per our response to reviewer 1, school readiness is currently a major focus in the UK public policy context in which it is conceptualised as a dichotomous outcome (despite the

availability of continuous scores) [17]. The aim of this paper is to contribute to this debate by identifying potential areas for targeting of resources to improve school readiness in the children who are likely to need the most support.

We have dichotomised the outcome 'school readiness' firstly in line with UK policy, and secondly to allow the testing of a PRM using ROC analysis which requires a binary outcome [18]. As in clinical prediction models which frequently have a binary outcome, outcomes have also been dichotomised in other predictive risk models looking at early cognitive development we are aware of e.g. [1–4]. A recent paper has also been published using MCS data which dichotomises outcomes for 3 child health and development indicators [19].

We acknowledge the limitations of dichotomising school readiness both conceptually (e.g. children develop at different rates) and statistically (i.e. loss of information). Further text has been added in the methods section, and we have acknowledged potential limitations in the discussion.

“We have dichotomised the outcome 'school readiness' in line with UK policy, and to allow the testing of a PRM using ROC analysis which requires a binary outcome [26].” p5, methods

“Many variables were dichotomised or grouped, which may be less sensitive than continuous measures.” p13, discussion

3.3. Comment: The statistical methods employed are varied and sophisticated. However, there are several errors and omissions in reporting, and in several places it is not possible to ascertain what exactly the authors did. In other places analytical decisions need to be better justified, where currently a justification appears to be lacking or to be incomplete. Please see below for specific comments on these issues.

Response: Thank you, we have addressed the specific issues raised in the comments below.

3.4. Comment: In the interest of Open Science, I strongly encourage the authors to make publicly available the Stata do-files they used to construct their data set and conduct their analysis. This would also help to clarify which specific procedures the authors used (e.g. in bootstrap optimism correction, analysis with weighted data, etc.).

Response: Thank you for highlighting this, we have made the Stata do file publicly available at: <https://www.dropbox.com/s/zxsl4cl87imydp0/SchoolreadinessPRM.do?dl=0>

This link is included in a supplementary file.

3.5. Comment: Title: The title should make clear that this study reports on the development of a risk model (without external validation), as recommended by #1 of the Tripod checklist. For example: “Development of a predictive risk model for school readiness at age 3 years using the UK Millennium Cohort Study”.

Response: We have changed the title as suggested.

“Development of a predictive risk model for school readiness at age 3 years using the UK Millennium Cohort Study”.p1

3.6. Comment: Strengths and Limitations: Having “a wide range of predictor variables” does not minimise the likelihood of overfitting. I suggest the authors reconsider which true strengths of their study they wish to highlight.

Response: Thank you for raising this point. We have reconsidered this point in the strengths and limitation box and reworded the first point as follows:

“Use of a large, representative, and contemporary cohort study to demonstrate the feasibility of predicting school readiness from data collected in infancy.”p3, text box

3.7. Comment: Methods: p.3 line 54: "Survey weightings were used to correct for attrition and non-response". (1) As it stands, this statement appears to be incomplete, as it does not mention weighting for sampling design. I suggest to move this sentence and integrate it with the first paragraph of the Statistical Analysis section, which appears to be more accurate (see also comments on p. 5 line 49). (2) I advise to refer to "weights", not weightings.

Response: Thank you for highlighting this. We have removed the sentence on survey weights from the p3 of the methods section. All information about survey weights has been consolidated in the statistical analysis section, and is referred to as weights not weightings.

"Survey weights were applied to take account of clustering, stratification and oversampling in the survey design, and attrition between survey waves, using the svyset command (pweight=BOVWT2) and svy prefix for regression modelling[37]."p6, methods

3.8. Comment: p. 4 line 3: Please provide a rationale why only singleton children were included in the study, and consider whether this is likely to affect the generalisability of the findings.

Response: Twins and triplets were excluded to ensure independence of observations, an assumption of logistic regression. Being from a multiple birth could be a predictor of school readiness, which was not tested in this model. If this is an important independent predictor, then the results would be less generalisable to the multiple birth cohort of children. We have acknowledged in the discussion that there are other potential predictors which could have been included in this model.

"...there are other predictors which may be associated with the outcome which were not included in this model..."p12, discussion

3.9. Comment: Outcome - It is not clear to me how and why the cut-off point (85) for dichotomising BSRA was chosen. Reference [22] suggests a categorisation, but without justification. Reference [23] does not appear to give any specific justification for dichotomising, or choosing a specific cut-off point. Please clarify on what basis the choice of cut-off point was made.

Response: The cut-offs recommended by Bracken reflected a 'normative classification' whereby children were categorised as very delayed, delayed, average, advanced or very advanced[10]. We used the same cut off score as Bracken (mean standardised composite score [MSCS] <85, which is 1 standard deviation below the mean) but collapsed the categories of delayed or very delayed into a single category equivalent to not being school ready. We have clarified this in the methods section.

"The BSRA raw scores were summed and adjusted for age to provide a standardised composite score[21]. Scores were grouped according to cut-offs recommended by Bracken which reflected a 'normative classification' whereby children were categorised as very delayed, delayed, average, advanced or very advanced [3]. We used the same cut off score as Bracken (mean standardised composite score <85, 1 standard deviation below mean) but collapsed the categories of delayed or very delayed into a single category equivalent to not being school ready."p4, methods

3.10. Comment: In fact, the cut-off point seems to lack validity, based on the figures reported by the authors: according to the authors' classification, 11.7 % of children were classified as 'not school ready' (p. 8), but Public Health England report that 31 % of children were regarded as 'not school ready' (p. 3) – almost three times as large a proportion. Given this apparent contradiction, I think the authors need to explain why they believe their cut-off point results in a valid indicator of school readiness.

Response: The Bracken scores were standardised in a US population, and have been validated as a measure of school readiness [6]. The reported figure of 31% of children 'not school ready' is based on administrative data from English schools. This measure is based on a teacher-led assessment of achievement against the learning goals of the Early Years Foundation Stage (EYFS) curriculum. The EYFS measure of school readiness is different from the Bracken School Readiness Assessment (BSRA) however there is evidence that they are strongly associated. An analysis of MCS data linked

with the EYFS outcomes showed that Bracken scores aged 3 were the most strongly associated variable with EYFS performance age 5 (OR for being in the bottom decile of EYFS scores: OR=5.8 'very delayed' BSRA score, OR=3.3 'delayed' BSRA, OR=0.4 'above average' BSRA score) [9]. We believe the cut off point is valid and have added some further information in the discussion:

“The main outcome, the BSRA, whilst validated as a measure of school readiness, was developed in the US and is not routinely used in the UK[23]. The BSRA measures a small set of pre-academic skills, but an analysis of MCS data linked to teacher reports showed that Bracken scores are strongly associated with the EYFS measure of school readiness used in English schools [4].”p13, discussion

3.11. Comment: Many continuous predictors were categorised, without justification. It seems to me that this has the potential to weaken the statistical model. Moreover, the cut-off points for categorisation appear arbitrary, since no justification is given for them. Categorisation often leads to loss of information (see the references to Altman and Senn in the general comments above). Why not use the full information from the predictor variables?

Response: Thank you for making this point, which has also been raised by other reviewers. We have conducted sensitivity analyses on the final model which is reported in supplementary file 1:

“As a sensitivity analysis the coding of 4 predictor variables was altered: maternal age (from categorical to continuous), developmental scores (from categorical to continuous) and ethnicity (from categorical to binary). The impact of this on final model performance is shown below.” SF1

Description	n	AUROC	Comparative AUROC (n=9310)
Final model	9487	0.80	0.79 (0.77,0.81)
Developmental score (continuous)	9487	0.80	0.80 (0.78,0.81)
Maternal age (continuous)	9310	0.79	0.79 (0.78,0.81)
Ethnicity (binary)	9487	0.79	0.79 (0.78,0.80)

Ho: area(xb1) = area(xb2) = area(xb3) = area(xb4); chi2(3) = 9.98; Prob>chi2 = 0.02

In summary, there were small but statistically significant differences between the models. The only change which improved model discrimination was using continuous development scores, so this was incorporated into the final model. There is a U-shaped relationship between school readiness and maternal age, so there was a clear rationale for including this as a categorical predictor.

3.12. Comment: p.5, line 33: Why was only mothers' social class considered as a predictor, and not fathers'?

Response: Social class was assigned according to the National Statistics Socio-Economic Classification (NS-SEC) by trained coders using answers given to questions about job details for main respondents. In the majority of cases, the Main interview was undertaken by the mother [20]. Data from the partner interviews is much less complete. This sentence has been clarified in the methods section:

“The National Statistics Socio-Economic Classification (NS-SEC) was used to code job details for main respondents (the majority of which were mothers) as:”p6, methods

3.13. Comment: Statistical analysis p.5, line 49: Survey weights. Please state the specific Stata command(s) used to weight the data and calculate correct standard errors for statistical analyses on weighted data (e.g. svyset, svy prefix). Please state the name of the weight variable from the MCS data set that was used.

Response: The Stata code used to set the survey weights is:

```
drop if BOVWT2==-1 svyset, clear
svyset [pweight=BOVWT2], strata(PTTYPE2) psu(SPTN00) fpc(NH2)
```

The svy prefix was used for regression modelling, but not to calculate the AUROC as it is not possible to use svy with these functions. The full code is now available via the dropbox link. We have expanded the description of survey weights in the statistical analysis section: "Survey weights were applied to take account of clustering, stratification and oversampling in the survey design, and attrition between survey waves, using the svyset command (pweight=BOVWT2) and svy prefix for regression modelling[37]."p6, methods

3.14. Comment: p. 5 line 52ff. Based on Table 1, the number of children not school ready in the complete cases sample is about $.117 \times 9487 = 1110$. This suggests an EPV of $1110/29 = 38$, not 68 as given in the manuscript. I see no reason to assume that the analysis presented here suffers from sparseness due to too many covariates, or from insufficient EPV. However, the following methodological points are worth considering: In support of their application of the EPV>10 rule of thumb, the authors cite reference Peduzzi et al (1995), which deals with EPV in proportional hazards regression, not logistic regression. I suspect they instead meant to cite: Peduzzi, Peter, et al. "A simulation study of the number of events per variable in logistic regression analysis." *Journal of clinical epidemiology* 49.12 (1996): 1373-1379. More importantly, the EPV>10 rule has not got a good evidence base, despite Peduzzi et al (1996), which reports on simulations performed on a single data set. I suggest to consider in addition the following, more recent reference, and to revise the manuscript accordingly: Courvoisier, Delphine S., et al. "Performance of logistic regression modeling: beyond the number of events per variable, the role of data structure." *Journal of clinical epidemiology* 64.9 (2011): 993-1000.

Response: Thank you for making these helpful points. The EPV was initially calculated based on the whole sample, not the completed cases which was the reason for the difference. This, and the incorrect Peduzzi reference, have been corrected. A full assessment of the performance of the model based on Courvoisier et al's paper is outside the scope of this analysis, however we have made the following amendment to 'go beyond' EPV in our assessment of model performance:

"The number of events per variable (EPV) exceeds 35, the predictors were checked for collinearity, a large number of predictors were used and all were significantly associated with the outcome suggesting a robust logistic regression model with sufficient sample size [37,38]." p6, methods

3.15. Comment: p.6, line 17: I suggest the authors consider also reporting the IDI_events and IDI_nonevents (see Pickering JW & Endre ZH (2012) New metrics for assessing diagnostic potential of candidate biomarkers. *Clinical Journal of the American Society of Nephrology* Vol. 7.

Response: Thank you for this suggestion. The Stata 'idi' command was used for the IDI analysis, and IDI_events and IDI_nonevents are not included in this output. We have used another Stata function to obtain the event and non-event data as shown below. However, as the IDI analysis is not a key message of this paper, and the word count is limited, we have not included this additional analysis in the manuscript.

Integrated Discrimination Improvement (and 95% b.s. CI's)

event IDI: 0.010 (0.006, 0.018)	non-event
IDI: 0.001 (0.001, 0.002)	
IDI: 0.012 (0.007, 0.020)	

3.16. Comment: p.6, line 28: Please clarify which method of optimism correction was employed. The cited reference [39] mentions three alternative methods. Also, more detail is required on the bootstrap: what was the number of bootstrap samples drawn? Finally, the "optimised" (optimism-corrected?) AUROC is not 'the difference between the baseline model performance and the performance across

the bootstrap samples', as the authors write. A clear short description of bootstrap optimism correction is given in: Austin & Steyerberg (2017) Events per variable and the relative performance of different strategies for estimating the out of sample validity of logistic regression models. *Statistical Methods in Medical Research* 26 (2): 796-808.

Response: Thank you for your helpful comments on the bootstrapping analysis. The Stata code used for this analysis is now shared as recommended so that readers can follow the code if interested. For the optimism correction we used bootstrapping, and 1000 bootstrap samples were drawn. We have added a reference to the Austin & Steyerberg (2017) paper and have used their definition of AUROC. We have amended the description of this analysis in line with your suggestions, as follows:

“Bootstrapping was used for internal validation; model performance was assessed using 1000 bootstrap samples, model optimism was averaged across all iterations to obtain an optimism estimate. An optimism-corrected AUROC, which takes account of overfitting, was calculated as the difference between unadjusted performance and the optimism estimate [21].”p7, results

3.17. Comment: p. 6, line 16: While the dominance analysis is sufficiently described, it is not clear to me how the authors decided to select 6 predictors for Model 2 from the 13 contained in Model 1. Why 6 and not 5 or 7, say? What criteria were used precisely? The results section (p. 10) states that IDI was used to assist in selecting the top 6 predictors, but it's not clear from the text or from Supplementary File 3 how the decision was made.

Response: Thank you for this comment, and apologies that this was not clear from the manuscript. The dominance analysis was used to rank the 13 predictors from the main model, IDI was used to assess the relative performance of model with 1,2,3...etc predictors included, started from the highest ranked predictors and adding in the subsequent predictor for each successive iteration.

Based on the results of the IDI, the most parsimonious model (model with less predictors) that gave us the best discrimination results compared with the saturated model (13 predictors) was chosen. 6 predictors offered improved discrimination over 5 but adding a 7th predictor had minimal impact. No formal criteria were used for this decision beyond the IDI analysis. The graph below shows IDI percentage for each iteration of the model. A 6-predictor model was chosen as we believe this offered the optimal balance between parsimony and discrimination.

Further information has been added to supplementary file 3:

“Integrated discrimination improvement (IDI) analysis was run using Stata function ‘idi’, which compares the discrimination ability between two logistic regression prediction models. In the first stage of this analysis, the IDI of a PRM with just the strongest

predictor variable (social class) was compared to a model with all 13 predictors. Adding the additional 12 predictors lead to a 7.3% increase in IDI. In each subsequent analysis, an additional predictor variable was added according to the ranking of variables from the dominance analysis (Table 1).” SF3

Predictor	Weighting	Rank
Social Class	17.38	1
Ethnic group	14.66	2
Maternal education	13.55	3
Income band	12	4
Gender	9.54	5
Number of children	7.84	6
Parent's employment	6.9	7
Housing type	5.65	8
Child development	3.9	9
Breastfeeding	3.9	10
Mother's age at birth of first child	2.87	11
Low birth weight	1.42	12
Mental health	0.38	13

Table 1 - Results of the dominance analysis for model 1

3.18. Comment: Page 6, line 39ff: The statement on ethical approval, while important, does not belong in the section on Statistical Analysis.

Response: Thank you for point this out. The ethics statement has been moved out of the statistical analysis section and included in the section with the PPI information:

“Ethics and Patient and public involvement

Ethical approval for each wave of the MCS was granted by NHS Multicentre Research Ethics Committees[44]. No further ethical approval was required for this secondary analysis of MCS data. There was no direct patient or public involvement in this analysis.

However the MCS has an ongoing programme of participant and public engagement.”p7, methods

3.19. Comment: Results - Model 2: The authors should clearly state from the beginning what the top 6 predictors are that are included in Model 2, before statistical comparisons of Model 1 and 2 are described (currently the six predictors are reported in brackets at the bottom of page 10). The authors should add a table displaying the estimated coefficients of Model 2. (Possibly this could be an additional column in Table 2.) An MI sensitivity analysis for Model 2 should be reported in a supplement.

Response: Thank you for these helpful points. The description of model 2 has been moved immediately after the dominance analysis (below table 1), with the 6 predictors described before the AUROC results are presented:

“IDI was used to test the relative performance of models with all (1-13) variables, with variables added in according to their rank from the dominance analysis (Supplementary File 3). These analyses informed the specification of model 2, which was comprised choice of the top 6- predictors: social class, child’s ethnic group, maternal education, income band, sex and number of children (see supplementary material 4 for Model 2 results).” P11, results

A table showing the coefficients of Model 2 for completed cases and an MI sensitivity analysis has been added in supplementary file 4.

3.20. Comment: MI data set: The sample size reported for the data set used for multiple imputation varies: it's given as 13,650 in Table 1, but 11,897 in Supplement 1.

Response: Thank you for highlighting this, we believe the reported figures are correct. The sample size for the full imputed data set is 13,650, this is the figure used in table 1 which is a demographic analysis of the sample. The sample size reported in supplement 1 is 11,897 because this is the number of cases included in the model. Not all variables were imputed e.g. the outcome school readiness, so there is still some missing data. The PRM uses data where all the predictor and outcome variables are available, so the sample size is less for this analysis.

3.21. Comment: p.10, line 41: Why does Model 2 have a different sample size to Model 1? IDI cannot be used to compare models based on different samples (p. 10 line 47).

Response: Thanks - you are right, and this is what we have done. The models were compared using the same sample size (n=9487), the complete case sample for the final model. We have made this clearer in the paper.

"The integrated discrimination improvement (IDI) using the complete case sample from model 1 was calculated to assess difference in performance between models as the percentage change in individuals being correctly assigned by the model[22]." p7, methods

3.22. Comment: p.10, line 42: Instead of 'bootstrap optimism', I suggest the term 'bootstrap optimism correction'.

Response: Thank you for this suggestion, we have made this change. The text now reads:

"Internal validation using bootstrap optimism correction suggests that the model would have good discriminatory power in an independent sample (adjusted AUROC model 1 = 0.79, model 2=0.76). ..."p11, results

3.23. Comment: p. 10, line 44: It is not clear whether the adjusted AUROC reported here refers to Model 1 or Model 2. Give the statistic for both models.

Response: Apologies that this wasn't clear in the original manuscript, the figure reported was for model 1. We have now included the adjusted AUROC for model 2 as well (see response to 3.22 for amended text).

3.24. Comment: Discussion - p. 11 line 27: "...suggesting it is possible to predict school readiness at age 3 using just six variables from the perinatal period and early infancy." I think this conclusion overstates the findings. There are several limitations that I think should receive stronger weight in the conclusions:

- The outcome variable was a measure of school readiness based on an apparently arbitrary cut-off on the BSRA, not an assessment of the child's readiness for school at the point of entering school.
- The outcome variable identified 12 % of children as not school ready, although official data suggest that the actual proportion of 'not-school-ready' children is twice to three times as large. So there are questions about the validity of the outcome used in this study.
- Sensitivity and specificity of the model are rather low, suggesting that practical application of this model would seem to be difficult and subject to many errors of identification. The authors do already allude to some of the negative consequences this could have in practice.

Response: Thanks – we have picked up these issues in response to your and other reviewer's comments and have made changes to the manuscript to give these issues more weight. The following text has been added:

In relation to the cut off point for the outcome variable:

“Scores were grouped according to cut-offs recommended by Bracken which reflected a ‘normative classification’ whereby children were categorised as very delayed, delayed, average, advanced or very advanced [10]. We used the same cut off score as Bracken (mean standardised composite score <85, 1 standard deviation below mean) but collapsed the categories of delayed or very delayed into a single category equivalent to not being school ready. We have dichotomised the outcome ‘school readiness’ in line with UK policy, and to allow the testing of a PRM using ROC analysis which requires a binary outcome [18].” p4-5, methods

In relation to validity of the outcome measure:

“A sensitivity analysis using an alternative outcome measure (British Ability Scales, BAS), showed that the BSRA measure led to improved discrimination (AUROC = 0.79 (95% CI 0.78-0.81) for BAS; AUROC = 0.80 (95% CI 0.78-0.81) for BSRA, p=0.002). See supplementary file 1 for further details.” p12, results

“The main outcome, the BSRA, whilst validated as a measure of school readiness, was developed in the US and is not routinely used in the UK[23]. The BSRA measures a small set of pre-academic skills, but an analysis of MCS data linked to teacher reports showed that Bracken scores are strongly associated with the EYFS measure of school readiness used in English schools [4].”p12-13, discussion

In relation to sensitivity and specificity of the model:

“However, the overall accuracy of the model is 74%, so over 200 children would be incorrectly classified; this could lead to stigmatisation of families and unnecessary use of resources. Nelson et al (2016) comment that Early Intervention services would be overwhelmed by the level of demand generated by such PRMs[18].” p14, discussion

3.25. Comment: p. 11 line 35: “risk factors of ECD”. Did the authors mean to say “risk factors of delayed cognitive development”?

Response: Thank you for pointing out this omission. We have amended this sentence as suggested:

“...have been identified as significant risk factors of delayed ECD...”p12, discussion

3.26. Comment: p. 12 line 9: The authors make too strong a claim regarding what internal validation via bootstrap can demonstrate. Bootstrapping cannot demonstrate generalisability to a different population. We can use the results from bootstrapping to evaluate how likely the results are to be replicated in another sample drawn in the same way from the same population.

Response: Thank you for this correction, we have revised this sentence to remove the comment about generalisability.

“Bootstrapping showed good internal validity suggesting the model would be generalisable to another population[48].”p12, discussion

3.27. Comment: p. 12 line 15: “Many variables were dichotomised or grouped ...”. See my comment under Methods. Since the authors are aware that this is a weakness, why did they choose to go down this route?

Response: We understand your concern about this point. Many of the predictor variables are categorical, others were grouped for ease of interpretation. Whilst we are aware this could reduce the

sensitivity of the data, there is also evidence that dichotomising a continuous variable can lead to a better predictive risk model[22]; in our own sensitivity analyses there was negligible impact of categorising predictor variables:

“As a sensitivity analysis the coding of 4 predictor variables was altered: maternal age (from categorical to continuous), developmental scores (from categorical to continuous) and ethnicity (from categorical to binary). The impact of this on final model performance is shown below:” SF4

Description	n	AUROC	Comparative AUROC (n=9310)
Final model	9487	0.80	0.79 (0.77,0.81)
Developmental (continuous) score	9487	0.80	0.80 (0.78,0.81)
Maternal age (continuous)	9310	0.79	0.79 (0.78,0.81)
Ethnicity (binary)	9487	0.79	0.79 (0.78,0.80)

Ho: $area(xb1) = area(xb2) = area(xb3) = area(xb4)$; $\chi^2(3) = 9.98$; $Prob > \chi^2 = 0.02$

In summary, there were small but statistically significant differences between the models. The only change which improved model discrimination was using continuous development scores, so this was incorporated into the final model. There is a U-shaped relationship between school readiness and maternal age, so there was a clear rationale for including this as a categorical predictor.

3.28. Comment: p. 12, line 48: An average English Local Authority with a population of 230,000 would therefore have 900 ‘at risk’ children per year.” It is not clear how this follows – the authors should describe how they made this calculation.

Response: Sorry that this was not made clear in the manuscript. The exemplar local authority had a population of 230,000, of which 3000 were under 1 year old. At the chosen cut off point ($p=0.12$) the model would result in a positive screen for 31% of the population. Assuming the model was applied at around age 1, approximately 900 children (31% of under 1 population) would be identified each year as being at risk of delayed school readiness. The text has been updated as follows:

“The model reported here would identify 31% of children screened as being ‘at risk’ of delayed school readiness. An exemplar English Local Authority with a total population of 230,000, and 3000 children aged under 1 year would identify 900 ‘at risk’ children per year if the PRM was applied to this cohort.”p13, discussion

3.29. Comment: Policy implications - The strongest predictors appear to be measures of the parents’ social status: occupational class, ethnicity, income, and education. Were this risk model to be applied in practice, could this mean that children from poorer, less educated, and ethnic minority families would be in danger of being stigmatised as being at risk of ‘not being school ready’? I think the potential social consequences of doing this need to be addressed in the discussion.

Response: Thank you for raising this issue, we agree that stigmatisation is a potential issue. This was mentioned in the discussion, but we have expanded this and made specific reference to the findings of this PRM:

“The model reported here would identify 31% of children screened as being ‘at risk’ of delayed school readiness. An exemplar English Local Authority with a total population of 230,000, and 3000 children aged under 1 year would identify 900 ‘at risk’ children per year if the PRM was applied to this cohort. This percentage equates with national data; in 2015/16,

31% of children in England were not school ready when tested at age 4-5[11]. However, the overall accuracy of the model is 74%, so over 200 children would be incorrectly classified; this could lead to stigmatisation of families and unnecessary use of resources.”p14, discussion

“PRMs raise ethical issues; labelling very young children as being at risk of poor development could be stigmatising for families, particularly when social factors are the strongest predictors as in this analysis. PRMs would generate false positives (and false negatives), which could cause unnecessary distress.”p14, discussion

3.30. Comment: Reference 19: The author’s last name is Hansen.

Response: Thank you for pointing out this error which has now been corrected (this is reference 21 in the revised manuscript).

“22 Kirstine Hansen. Millennium Cohort Study. First, Second, Third and Fourth Surveys. A Guide to the Datasets. Centre for Longitudinal Studies 2012. <http://www.cls.ioe.ac.uk/shared/get-file.ashx?id=598&itemtype=document>” references

3.31. Comment: Reference 22: The URL does not link to the document.

Response: Thank you for pointing this out. We have removed the URL from the reference.

3.32. Comment: Supplementary File 2 - Please use the same reference category for “gender” in the main manuscript and in the MI analysis.

Response: Thank you for this suggestion, we have changed the reference category to ‘female’ in the MI analysis as it was in the main analysis. The updated table for supplementary file 2 is shown below, this also includes development scores as a continuous variable, and the results of the dominance analysis for the MI data in model 1:

Predictor	Adjusted OR (95% CI)	Weighting (rank)
GROUP 1 - DEMOGRAPHIC & INDIVIDUAL FACTORS		
Gender		8.5 (5)
Female	1	
Male	1.86 (1.62,2.14)	
Ethnicity		15.7 (3)
White	1	
Mixed	1.04 (0.62,1.75)	
Indian	2.68 (1.85,3.89)	
Pakistani and Bangladeshi	3.85 (2.94,5.04)	
Black or Black British	2.31 (1.43,3.72)	
Other ethnic group	3.95 (2.30,6.77)	
Mother's age at birth of first child		1.5 (12)
30-39	1	
40+	1.05 (0.67,1.64)	
20-29	1.22 (0.99,1.51)	
14-19	1.22 (0.93,1.59)	
Birth weight (<2500grams)		

Normal/high	1	
Low birthweight	1.52 (1.18,1.97)	1.2 (13)
Maternal Mental Health (Diagnosed depression/anxiety)		
No	1	
Yes	1.15 (0.98,1.34)	1.5 (11)
Child developmental milestones		
Developmental score	1.10 (1.07,1.13)	2.8 (10)
GROUP 2 - LIFESTYLE FACTORS		
Duration of breastfeeding		
6 months or more	1	3.6 (9)
6 weeks - 6 months	1.17 (0.92,1.48)	
One week or less	1.15 (0.90,1.48)	
1 - 6 weeks	1.22 (0.96,1.57)	
Never	1.58 (1.29,1.95)	
GROUP 3 - SOCIAL & COMMUNITY NETWORKS		
Number of children in family		
One child	1	7.1 (6)
Two or three children	1.40 (1.19,1.63)	
Four or more children	2.48 (1.94,3.16)	
GROUP 4 - LIVING & WORKING CONDITIONS		
Maternal education		
Degree plus	1	16.7 (2)
Diploma	0.88 (0.61,1.26)	
A levels	1.13 (0.80,1.59)	
GCSE A-C	1.34 (1.01,1.78)	
GCSE D-G	1.72 (1.23,2.39)	
None	1.74 (1.28,2.38)	
Workforce status		
Both parents in work	1	6.5 (7)
One parent in work	0.94 (0.78,1.12)	
Neither parent in work	1.21 (0.93,1.57)	
Housing tenure		
Owner occupied	1	5.5 (8)
Private rented	1.18 (0.90,1.54)	
Social housing	1.43 (1.18,1.72)	
Other	0.96 (0.69,1.35)	
GROUP 5 - SOCIOECONOMIC AND WIDER FACTORS		
Social class		
Managerial & professional	1	
Intermediate	0.98 (0.75,1.29)	

Small employers & own account	1.32 (0.87,2.00)	17.6 (1)
Lower supervisory & technical	1.50 (1.06,2.13)	
Semi-routine & routine	1.77 (1.38,2.27)	
Never worked & long-term unemployed	2.19 (1.53,3.15)	
Annual income		11.9 (4)
£33,000+	1	
£22,000-£33,000	1.33 (1.02,1.72)	
£11,000-£22,000	1.67 (1.30,2.14)	
£0-£11,000	2.14 (1.60,2.87)	
ROC Analysis	AUROC = 0.79 (95% CI 0.78,0.80)	

3.33. Comment: Supplementary File 3 - I don't understand what this table is showing me. Is each model compared to the previous one? It would be helpful to have a description of exactly what was done, and which variables are contained in each model. Since there are 13 candidate predictors, wouldn't the reader expect there to be 13 models to be assessed? Also, I'm not sure how this table helps to select 6 predictors for Model 2 (see also comment on p. 6 line 16).

Response: Apologies that this information wasn't presented clearly enough. Each line of the table is a comparison between a model containing x number of variables and the full model with 13 variables. As at least 1 variable is required for comparison, this is why there are 12 models instead of 13. This information has been added to supplementary file 3 to give a more detailed explanation. Further information on why 6 predictors were selected is included in response 3.17.

"Integrated discrimination improvement (IDI) analysis was run using Stata function 'idi', which compares the discrimination ability between two logistic regression prediction models. In the first stage of this analysis, the IDI of a PRM with just the strongest predictor variable (social class) was compared to a model with all 13 predictors. Adding the additional 12 predictors lead to a 7.3% increase in IDI. In each subsequent analysis, an additional predictor variable was added according to the ranking of variables from the dominance analysis (Table 1)." SF3

Predictor	Weighting	Rank
Social Class	17.38	1
Ethnic group	14.66	2
Maternal education	13.55	3
Income band	12	4
Gender	9.54	5
Number of children	7.84	6
Parent's employment	6.9	7
Housing type	5.65	8
Child development	3.9	9
Breastfeeding	3.9	10
Mother's age at birth of first child	2.87	11

Low birth weight	1.42	12
Mental health	0.38	13

VERSION 2 – REVIEW

REVIEWER	Sharon Wolf University of Pennsylvania, USA
REVIEW RETURNED	18-Feb-2019

GENERAL COMMENTS	Overall, the authors have been very responsive to the reviews and have addressed my comments sufficiently. I understand their constraints in terms of the word limits of the journal, and thus believe they have done what they can to address the reviews from all of the reviewers. I have only one additional comment, which relates to #1.1. While the authors make the case for why they define the BSRA pre-academic skills as “school readiness”, it should at least be acknowledged and listed as a limitation of the study that this is a very limited measure of school readiness, which is usually defined as a broad set of skills that include behavioral, social, and cognitive skills. Particularly given the very small correlations with the other measures of child development (0.06), this concern is even greater. Please add this as a limitation.
--

REVIEWER	Orla Doyle University College Dublin, Ireland
REVIEW RETURNED	08-Feb-2019

GENERAL COMMENTS	I would like to thank the authors for addressing all my comments. The new version of the manuscript is much improved. I just have 2 points:  1. One of my comments referred to research showing that parenting skills and the quality of the home environment are significantly associated with children's cognitive skills. And that measures of these are available in MSC. While the author have now included a measure of childcare at age 9 as a potential predictor, they did not address my comment regarding parental and HOME scores. Could the authors please explain why these measures are not considered? 2. Page 5 and 6 describe the 5 groups of predictor variables considered. However the variables described in each group do not always correspond to the variables included in the groups in Table 1. For example, maternal education and housing tenure are described as part of Group 4 Living and Working Conditions, but in Table 1 maternal educated is listed under Social and Community Networks and housing tenure is listed under Group 5 Socioeconomic and wider factors. Also, I suggest changing the name of group 3 'Social and Community factors' as it does not include any community factors....these are listed under Group group 4 (question and pollution). Also, should maternal attachment really be listed under Living and Working Conditions?
---

REVIEWER	Peter Martin Department of Applied Health Research University College London United Kingdom
REVIEW RETURNED	28-Feb-2019

GENERAL COMMENTS	I thank the authors for the careful way in which they have addressed the comments of all reviewers. They have responded thoughtfully, meticulously and sagaciously, and for the most part have made appropriate changes that in my opinion have improved the manuscript. I am particularly grateful that the authors have made their Stata code available. This improves the reproducibility of their study, but I hope it will also encourage others to follow suit and help make science more open. There are four issues on which I am not persuaded that my points have been completely addressed, or on which I have further queries. I will set these out first, before listing some smaller points. Validity of the BSRA All three reviewers have raised concerns regarding the validity of the BSRA as a measure of school readiness. I recognise that the authors have made helpful changes to their manuscript to emphasise both the strengths and the limitations of the BSRA as a measure of school readiness. Among other things, the authors cite Kiernan & Hobcraft (2010), who report on a study in which MCS data were linked to teacher's assessments of school readiness ("FSP score"). Could the authors use this linked data set to test how well their model predicts teacher's assessments, and would such an analysis provide further validation of their model? And why did the authors choose the BSRA as the outcome for their study in the first place, rather than the FSP score, since the latter is linked to MCS and available? In the Discussion, the authors prudently point to the need for external validation of their model in a new data set. I wonder if the authors would additionally recommend validating their prediction model in a new data set with teachers' assessments of school readiness as the outcome? Dichotomous versus continuous outcome Two reviewers questioned why the Bracken score was dichotomised for the purpose of this paper. The authors have chosen to stand by their choice, and have justified it in two ways: (1) This is how the concept is discussed in policy circles, and (2) it allowed calculation of the AUROC. Ad (1): Isn't it (also) the job of science to develop and investigate concepts, and then communicate them to non-scientists, including policy makers – rather than assume concepts from policy as given and develop science around them? Ad (2): ROC curve analysis is only useful if the outcome is dichotomous, but it would be perfectly possible to develop, evaluate and internally validate a model for school readiness measured on an interval scale (or on the ordered categorical scale recommended by Bracken, for that matter). Findings from such models also have the potential to be used to inform policy, with the difference that recommendations would have to be more nuanced than classifying children as either 'school-ready' or not. In short, I remain unconvinced by the authors' stated reasons for dichotomisation. However, I don't wish this objection to prevent publication of this manuscript. I recommend instead to explicitly
--

mention the dichotomisation of the outcome as a limitation in the Discussion, as this may stimulate debate around methods used in this area, and may inspire other scientists to try different methods. The authors may also consider citing one or both of the references on the disadvantages of dichotomisation that I recommended in my first review.

Optimism correction via bootstrapping:

The explanation of the optimism correction is now clearer than it was, but an important point remains unclear to me. Did the authors repeat all the modelling steps (including selection of predictors) in each bootstrap sample, as recommended by the reference they cite (Austin & Steyerberg 2017, p. 799)? If I read their Stata code correctly, they did not. This should be clarified and justified.

Multiple Imputation (MI)

I believe there remains an inconsistency in the sample sizes the authors report for the MI data set. I am not convinced by their reply to my previous comment on this (response letter page 23, point 3.20). Naturally it does not make sense to impute the outcome. The first sentence of the results section states that 13,650 children had the outcome recorded, and the same number is reported as the size of the "imputed sample" in Table 1. The authors have not given a reason for excluding any of the covariates from the imputation model. So the sample size for the sensitivity analysis with multiply imputed data ought also to equal 13,650, as far as I can see (rather than 11,879 as given in Supplement 2).

p. 3 line 35: "School readiness is currently a major focus in England".

Please specify for whom exactly this is a major focus.

p. 3 line 37 ff: "There was nearly a 20% point gap in performance between the most (62% school ready) and the least (80%) deprived deciles of Index of Multiple Deprivation".

I understand what the authors mean, but the phrasing is potentially confusing. Please consider revising. For example: "...between children living in the most deprived areas (62 % school ready) and children living in the least deprived areas (80 % school ready), when areas are classified into deciles according to the Index for Multiple Deprivation."

p. 5 line 30f: "Indices of Multiple Deprivation (IMD) from 2004 were linked retrospectively to wave 1 data to give small area level deprivation measure."

The formulation may give the misleading impression that the authors did the linkage. Please rephrase and cite the relevant MCS documentation.

p. 7, lines 34ff: "An optimism-corrected AUROC, which takes account of overfitting, was calculated as the difference between unadjusted performance and the optimism estimate".

This is not wrong, but formulated in an unnecessarily unclear way. The optimism-corrected AUROC is calculated by subtracting the optimism estimate from the uncorrected AUROC.

p. 10, Table 2: OR for Developmental Score

Would it be better to standardise the variable “Development Score”, so that the OR gives the effect of a 1 SD change in the log odds of the outcome? (Equivalently, the authors could x-standardise the coefficient of the logistic regression.) Currently the OR gives the effect of a 1-point change in “Development Score”, but I don’t know what a 1-point change in “Development Score” means. I note that I’m not an expert in this area of measurement. If experts find it easier to interpret a 1-point change on this measure rather than a 1 SD change, then it’s better to keep things as they are.

P 12 lines 5f: “At a probability cut-off of 12%, 31% of the screened population tested would be identified as being at high risk of poor school readiness using model 1.”

I think it would be better to call the children thus identified ‘at risk’, rather than ‘at high risk’. This change would be consistent with changes made elsewhere in the manuscript.

p.12 lines 29ff: “A parsimonious model performed similarly well (AUROC=0.78), suggesting it is possible to predict school readiness at age 3 using just six variables from the perinatal period and early infancy”.

I appreciate the authors’ response to my original comment on this sentence, and think they have made thoughtful changes to improve the manuscript. However, I still think that this sentence carries the risk of being misinterpreted, such that a reader may get too rosy a view of the model’s predictive performance. I suggest to amend to “... it is possible to predict school readiness at age 3 fairly well using just six variables ...”. This would be consistent with the authors’ own classification of $.7 < \text{AUROC} < .8$ as ‘fair’.

p. 13 lines 23ff: “Many variables were dichotomised or grouped, which may be less sensitive than continuous measures”.

Maybe the authors could consider revising this sentence to be more specific and incisive, in line with my comment on the limitations of dichotomising the outcome.

p. 14 lines 8ff: However, the overall accuracy of the model is 74%, so over 200 children would be incorrectly classified; this could lead to stigmatisation of families and unnecessary use of resources. Correct classification can also lead to stigmatisation. The authors’ discussion of stigmatisation risk in the paragraph that follows is more appropriate in this respect, in my opinion.

Reference 4 (Hobcraft & Kiernan):

The URL does not link to the document.

Supplementary File 1, categorising age at first child:

The authors state that the apparent U-shaped relationship between Mother’s Age at first child and the risk of not being school ready justifies categorising the age variable. Have the authors considered a quadratic transformation of age instead?

VERSION 2 – AUTHOR RESPONSE

1. Reviewer 1 - Sharon Wolf

1.1. Comment: Overall, the authors have been very responsive to the reviews and have addressed my comments sufficiently. I understand their constraints in terms of the word limits of the journal, and thus believe they have done what they can to address the reviews from all of the reviewers.

Response: Thank you, we endeavoured to respond to all the comments as fully as possible.

1.2. Comment: I have only one additional comment, which relates to #1.1. While the authors make the case for why they define the BSRA pre-academic skills as “school readiness”, it should at least be acknowledged and listed as a limitation of the study that this is a very limited measure of school readiness, which is usually defined as a broad set of skills that include behavioral, social, and cognitive skills. Particularly given the very small correlations with the other measures of child development (0.06), this concern is even greater. Please add this as a limitation.

Response: We acknowledge this limitation of the BSRA and have included the below statement to the discussion to make this more explicit:

“The BSRA measures a small set of pre-academic skills, and as such is a limited measure of child development, which can be defined as including broader behavioural and social skills.” p13, discussion

2. Reviewer 2 – Orla Doyle

2.1. Comment: I would like to thank the authors for addressing all my comments. The new version of the manuscript is much improved.

Response: Thank you, we agree that responding to the comments has been a valuable process.

2.2. Comment: One of my comments referred to research showing that parenting skills and the quality of the home environment are significantly associated with children's cognitive skills. And that measures of these are available in MSC. While the authors have now included a measure of childcare at age 9 (months) as a potential predictor, they did not address my comment regarding parental and HOME scores. Could the authors please explain why these measures are not considered?

Response: We have reviewed the data captured in the first MCS wave at age 9 months again and are not aware of any equivalent measures for assessing parental investment and home learning environment. We identified ‘parenting activities’ questions which include how often a child is read to, but these were asked from MCS2 onwards, not in the first wave. The Home Observation for

Measurement of the Environment (HOME-SF) scale was included in MCS2 when children were aged 3, but also not in the first wave of data collection[1]. Parents in the first wave were asked about their parenting beliefs, which included questions on providing stimulation for babies. However, the purpose of these was to assess their attitudes to child-rearing, not to identify the frequency of these behaviours, so we do not regard this as an analogous measure. We have added a further limitation that home learning environment was not considered in this analysis.

“...there are other predictors which may be associated with the outcome which were not included in this model e.g. the home learning environment (which was not assessed at 9 months in the MCS) and childcare in infancy [50].” p13, discussion

2.3. Comment: Page 5 and 6 describe the 5 groups of predictor variables considered. However the variables described in each group do not always correspond to the variables included in the groups in Table 1. For example, maternal education and housing tenure are described as part of Group 4 Living and Working Conditions, but in Table 1 maternal educated is listed under Social and Community Networks and housing tenure is listed under Group 5 Socioeconomic and wider factors. Also, I suggest changing the name of group 3 'Social and Community factors' as it does not include any community factors....these are listed under Group group 4 (question and pollution). Also, should maternal attachment really be listed under Living and Working Conditions?

Response: Thank you for spotting this inconsistency. Maternal education and housing were included under the wrong headings in table 1. We have checked the rest of the manuscript and believe the groups are consistently represented now. We have changed the name of group 3 to community networks, not factors, and recognise that this will mainly represent family members for a small child. We accept that maternal attachment could also have been included in social networks, however we think this forms part of the child's 'living conditions' which was the justification for including it in group 4.

“Group 3 – Social and Community Networks” p5, methods

3. Reviewer 3 – Peter Martin

3.1. Comment: I thank the authors for the careful way in which they have addressed the comments of all reviewers. They have responded thoughtfully, meticulously and sagaciously, and for the most part have made appropriate changes that in my opinion have improved the manuscript. I am particularly grateful that the authors have made their Stata code available. This improves the reproducibility of their study, but I hope it will also encourage others to follow suit and help make science more open.

Response: Thank you for your time and diligence in providing comments which have improved the quality of this manuscript.

3.2. Comment: All three reviewers have raised concerns regarding the validity of the BSRA as a measure of school readiness. I recognise that the authors have made helpful changes to their manuscript to emphasise both the strengths and the limitations of the BSRA as a measure of school readiness. Among other things, the authors cite Kiernan & Hobcraft (2010), who report on a study in which MCS data were linked to teacher's assessments of school readiness ("FSP score"). Could the authors use this linked data set to test how well their model predicts teacher's assessments, and would such an analysis provide further validation of their model? And why did the authors choose the BSRA as the outcome for their study in the first place, rather than the FSP score, since the latter is linked to MCS and available?

Response: The FSP data is available as a linked education dataset which can be accessed via the UK data service, which requires secure access. This research was done as part of an MPH dissertation; we did not have access to the FSP data and it was not possible to gain access within the timeframe. FSP data is only available for children in England, where consent has been given, and linkage was possible, so the sample size would be reduced. However, we agree that the next stage of this work could involve validation of the model using the FSP scores from teacher's assessments.

"Further research is needed to test the external validity of predictive risk models for ECD for example in another cohort or with linked administrative datasets such as the EYFS data from English schools."
p14, discussion

3.3. Comment: In the Discussion, the authors prudently point to the need for external validation of their model in a new data set. I wonder if the authors would additionally recommend validating their prediction model in a new data set with teachers' assessments of school readiness as the outcome?

Response: Yes, we agree with this and have added this as a recommendation for further research (see response to comment 3.2 for additional text added to the manuscript).

3.4. Comment: Two reviewers questioned why the Bracken score was dichotomised for the purpose of this paper. The authors have chosen to stand by their choice, and have justified it in two ways: (1) This is how the concept is discussed in policy circles, and (2) it allowed calculation of the AUROC.

Ad (1): Isn't it (also) the job of science to develop and investigate concepts, and then communicate them to non-scientists, including policy makers – rather than assume concepts from policy as given and develop science around them?

Ad (2): ROC curve analysis is only useful if the outcome is dichotomous, but it would be perfectly possible to develop, evaluate and internally validate a model for school readiness measured on an interval scale (or on the ordered categorical scale recommended by Bracken, for that matter). Findings from such models also have the potential to be used to inform policy, with the difference that recommendations would have to be more nuanced than classifying children as either 'school-ready' or not.

In short, I remain unconvinced by the authors' stated reasons for dichotomisation. However, I don't wish this objection to prevent publication of this manuscript. I recommend instead to explicitly mention the dichotomisation of the outcome as a limitation in the Discussion, as this may stimulate debate around methods used in this area, and may inspire other scientists to try different methods. The authors may also consider citing one or both of the references on the disadvantages of dichotomisation that I recommended in my first review.

Response: We acknowledge your concerns about this methodological approach and have added some further text in the discussion to explicitly recognise the limitations of dichotomising the outcome (including the 2 references suggested previously), and have also added a suggestion for further research to explore other predictive modelling approaches.

"The outcome variable was dichotomised to allow ROC curve analysis. We acknowledge the limitations of dichotomising school readiness ethically, conceptually (e.g. children develop at different rates) and statistically (i.e. loss of information) [49,50]". p13, discussion

"Alternative modelling approaches which do not require a dichotomous outcome could also be tested. Findings from such models could offer more nuanced predictions on school readiness." p14, discussion

3.5. Comment: The explanation of the optimism correction is now clearer than it was, but an important point remains unclear to me. Did the authors repeat all the modelling steps (including selection of predictors) in each bootstrap sample, as recommended by the reference they cite (Austin & Steyerberg 2017, p. 799)? If I read their Stata code correctly, they did not. This should be clarified and justified.

Response: As you have correctly identified, the bootstrap analysis was used solely to obtain optimism correction for the AUROC estimate based on the variables already selected by stepwise selection; selection of the predictors in each bootstrap sample was not repeated. We have clarified this in the methods section.

"Bootstrapping was used for internal validation of the final models, without repeating selection of predictors in each bootstrap sample. Model performance was assessed using

1000 bootstrap samples, model optimism was averaged across all iterations to obtain an optimism estimate. An optimism-corrected AUROC, which takes account of overfitting,

was calculated by subtracting the optimism estimate from the uncorrected [42]." P7, methods

3.6. Comment: I believe there remains an inconsistency in the sample sizes the authors report for the MI data set. I am not convinced by their reply to my previous comment on this (response letter page 23, point 3.20). Naturally it does not make sense to impute the outcome. The first sentence of the

results section states that 13,650 children had the outcome recorded, and the same number is reported as the size of the “imputed sample” in Table 1. The authors have not given a reason for excluding any of the covariates from the imputation model. So the sample size for the sensitivity analysis with multiply imputed data ought also to equal 13,650, as far as I can see (rather than 11,879 as given in Supplement 2).

Response: The sample size for the imputed sample is 11,897, as reported in Supplement 2, and this has now been corrected in table 1. You are correct in pointing out that 13,650 children had a recorded outcome. The following covariates were used to shape the imputation process and so were not imputed: School readiness (outcome), maternal education, child’s gender and mother’s age at birth of first child. This is the reason why the imputed sample size is less than 13,650. Further detail has been added to the methods section to clarify this:

“As a sensitivity analysis, multiple imputation by chained equation was performed to impute missing data using the ‘mi impute chained’ command in Stata. Three predictor

variables from the first sweep (maternal education, child’s sex, mother’s age at birth of first child) and the outcome variable were to shape the imputation process (imputed sample, n=11,897). Twenty imputed datasets were generated, and Rubin’s rules were used to calculate results across the imputed datasets[43].” p7, methods

3.7. Comment: p. 3 line 35: “School readiness is currently a major focus in England”. Please specify for whom exactly this is a major focus.

Response: We have added some further detail to this sentence.

“School readiness is currently a major focus in England for policy makers, educators and the public health community...” p3, introduction

3.8. Comment: p. 3 line 37 ff: “There was nearly a 20% point gap in performance between the most (62% school ready) and the least (80%) deprived deciles of Index of Multiple Deprivation”. I understand what the authors mean, but the phrasing is potentially confusing. Please consider revising. For example: “...between children living in the most deprived areas (62 % school ready) and children living in the least deprived areas (80 % school ready), when areas are classified into deciles according to the Index for Multiple Deprivation.”

Response: Thank you for this suggestion. We have rephrased this sentence to try and remove ambiguity.

“The percent of children school ready was nearly 20% higher in the most affluent decile

(80% school ready) compared to the most deprived decile (62% school ready) when areas were classified into deciles according to the Index for Multiple Deprivation [12].” p3, introduction

3.9. Comment: p. 5 line 30f: “Indices of Multiple Deprivation (IMD) from 2004 were linked retrospectively to wave 1 data to give small area level deprivation measure.” The formulation may give the misleading impression that the authors did the linkage. Please rephrase and cite the relevant MCS documentation.

Response: This sentence has been rephased and the appropriate MCS documentation cited:

“Indices of Multiple Deprivation (IMD) from 2004 which had been retrospectively linked to wave 1 data were used to give small area level deprivation measures [20]. IMD scores were divided into quintiles, with 1 the most deprived quintile, and 5 the least deprived.” p6, methods

3.10. Comment: p. 7, lines 34ff: “An optimism-corrected AUROC, which takes account of overfitting, was calculated as the difference between unadjusted performance and the optimism estimate”. This is not wrong, but formulated in an unnecessarily unclear way. The optimism-corrected AUROC is calculated by subtracting the optimism estimate from the uncorrected AUROC.

Response: We have rephrased this sentence in line with this suggestion:

“An optimism-corrected AUROC, which takes account of overfitting, was calculated by subtracting the optimism estimate from the uncorrected AUROC [42].” P7, methods

3.11. Comment: p. 10, Table 2: OR for Developmental Score Would it be better to standardise the variable “Development Score”, so that the OR gives the effect of a 1 SD change in the log odds of the outcome? (Equivalently, the authors could x-standardise the coefficient of the logistic regression.) Currently the OR gives the effect of a 1-point change in “Development Score”, but I don’t know what a 1-point change in “Development Score” means. I note that I’m not an expert in this area of measurement. If experts find it easier to interpret a 1-point change on this measure rather than a 1 SD change, then it’s better to keep things as they are.

Response: This is an interesting point. It would be possible to use a standardised score, however we think it is more intuitive to leave it as a 1-point change. The measure of child development available in the MCS is a combination of items from the Denver

Developmental Screening Test and the MacArthur Communicative Development Inventory. We do not want to introduce confusion by reporting a standardised score.

3.12. Comment: P 12 lines 5f: "At a probability cut-off of 12%, 31% of the screened population tested would be identified as being at high risk of poor school readiness using model 1." I think it would be better to call the children thus identified 'at risk', rather than 'at high risk'. This change would be consistent with changes made elsewhere in the manuscript.

Response: Thank you for this suggestion, we have changed the text accordingly:

"At a probability cut-off of 12%, 31% of the screened population tested would be identified as being 'at risk' of poor school readiness using model 1." p12, results

3.13. Comment: p.12 lines 29ff: "A parsimonious model performed similarly well (AUROC=0.78), suggesting it is possible to predict school readiness at age 3 using just six variables from the perinatal period and early infancy". I appreciate the authors' response to my original comment on this sentence, and think they have made thoughtful changes to improve the manuscript. However, I still think that this sentence carries the risk of being misinterpreted, such that a reader may get too rosy a view of the model's predictive performance. I suggest to amend to "... it is possible to predict school readiness at age 3 fairly well using just six variables ...". This would be consistent with the authors' own classification of $.7 < \text{AUROC} < .8$ as 'fair'.

Response: Thank you for this suggestion. We have amended the text as you suggested:

"A parsimonious model performed similarly well (AUROC=0.78), suggesting it is possible to predict school readiness at age 3 fairly well using just six variables from the perinatal period and early infancy." p12, discussion

3.14. Comment: p. 13 lines 23ff: "Many variables were dichotomised or grouped, which may be less sensitive than continuous measures". Maybe the authors could consider revising this sentence to be more specific and incisive, in line with my comment on the limitations of dichotomising the outcome.

Response: As per the response to comment 3.4 we have amended this text to comment specifically on the limitation of dichotomising the outcome:

"The outcome variable was dichotomised to allow ROC curve analysis; there are disadvantages to this approach including potential loss of information [49,50]". p13, discussion

3.15. Comment: p. 14 lines 8ff: However, the overall accuracy of the model is 74%, so over 200 children would be incorrectly classified; this could lead to stigmatisation of families and unnecessary use of resources. Correct classification can also lead to stigmatisation. The authors' discussion of stigmatisation risk in the paragraph that follows is more appropriate in this respect, in my opinion.

Response: We agree there was some repetition in the discussion on stigmatisation. The sentence you identified has been amalgamated with the text in the following paragraph to bring this point together in the discussion:

“However, the overall accuracy of the model is 74%, so over 200 children would be incorrectly classified. PRMs raise ethical issues; labelling very young children as being at risk of poor development could be stigmatising for families, particularly when social factors are the strongest predictors as in this analysis. PRMs would generate false positives (and false negatives), which could cause unnecessary distress and use of resources.” p14, discussion

3.16. Comment: Reference 4 (Hobcraft & Kiernan): The URL does not link to the document.

Response: This URL has now been removed.

3.17. Comment: Supplementary File 1, categorising age at first child: The authors state that the apparent U-shaped relationship between Mother’s Age at first child and the risk of not being school ready justifies categorising the age variable. Have the authors considered a quadratic transformation of age instead?

Response: Thank you for this suggestion. We had not considered a quadratic transformation of this data. Whilst this is something which could be carried out in future work, we are satisfied from the sensitivity analysis reported in Supplementary File 1 that altering the coding of maternal age from categorical to continuous has negligible impact on the performance of the PRM. In this instance we do not believe that a quadratic transformation is warranted.

VERSION 3 - REVIEW

REVIEWER	Sharon Wolf University of Pennsylvania USA
REVIEW RETURNED	20-Apr-2019

GENERAL COMMENTS	The authors have addressed all of my concerns.
--

REVIEWER	Orla Doyle University College Dublin, Ireland
REVIEW RETURNED	10-Apr-2019

GENERAL COMMENTS	I would like to thank the authors for addressing my remaining questions. I am happy with their revisions.
---

REVIEWER	Peter Martin University College London, United Kingdom
REVIEW RETURNED	15-Apr-2019

GENERAL COMMENTS	Thanks to the author team for responding so carefully to the reviewers' comments once again. I think the additional changes have further improved the manuscript. There is just one point I'd like to return to. Multiple imputation. It is not clear to me what the authors mean by 'three predictor variables from the first sweep were [used to] to shape the imputation process', and why that explains the sample size of their imputed data sets. What does 'shape the imputation process' mean? If the three predictors (mother's education, child's gender and mother's age at first birth) were included in the imputation model (as they should be), then missing values on these variables should have been imputed also. Naturally, it does not make sense to impute the outcome.
--

VERSION 3 – AUTHOR RESPONSE

We are pleased to have addressed the comments from reviewers 1 and 2 and appreciate this opportunity to respond to a final comment from the reviewer 3, who has also recommended publication.

Reviewer 3 commented on the multiple imputation process, stating that, "It is not clear to me what the authors mean by 'three predictor variables from the first sweep were [used to] to shape the imputation process", and why that explains the sample size of their imputed data sets. What does 'shape the imputation process' mean? If the three predictors (mother's education, child's gender and mother's age at first birth) were included in the imputation model (as they should be), then missing values on these variables should have been imputed also. Naturally, it does not make sense to impute the outcome."

Response: Thanks, and apologies for any confusion. As the reviewer suggests, we used these predictor variables in the imputation model to generate the imputation dataset, as relatively little data were missing for early sweeps on these variables, giving an imputed sample of 11,897. We have now clarified this further in the manuscript:

"As a sensitivity analysis, multiple imputation by chained equation was performed to impute missing data using the 'mi impute chained' command in Stata. We used predictor variables with relatively little missing

data (maternal education, child's sex, mother's age at birth of first child) and the outcome as regular variables

in the imputation model. As such individuals with missing data for these 4 items were not included in the final imputed sample (n=11,897). Twenty imputed datasets were generated, and Rubin's rules were used to calculate results across the imputed datasets [43]." p7, methods